# Employing Copernicus Land Service and Sentinel-2 Satellite Mission Data to Assess the Spatial Dynamics and Distribution of the Extreme Forest Fires of 2023 in Greece

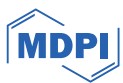

Anna Dosiou [1], Ioannis Athinelis [1], Efstratios Katris [1], Maria Vassalou [1], Alexandros Kyrkos [1], Pavlos Krassakis [1,2,*] and Issaak Parcharidis [1,*]

1    Department of Geography, Harokopio University of Athens, Eleftheriou Venizelou 70, 17676 Athens, Greece; gp222305@hua.gr (A.D.); gp222301@hua.gr (I.A.); gs21823@hua.gr (E.K.); gp221301@hua.gr (M.V.); gs21997@hua.gr (A.K.)

2    Centre for Research & Technology Hellas (CERTH), 15125 Athens, Greece

*    Correspondence: pkrassakis@hua.gr (P.K.); parchar@hua.gr (I.P.)

**Abstract:** In 2023, Greece faced its worst wildfire season, with nine major fires causing unprecedented environmental damage of 1470.31 km$^2$. This article uses Copernicus Land Monitoring Service and Sentinel-2 data, employing advanced remote sensing and GIS techniques to analyze spatial dynamics, map burn severity, assess fire extent, and highlight pre-fire tree density and land cover. The study focuses on the catastrophic fire in the Evros region and the damage to the National Forest Park of Dadia–Lefkimmi–Soufli. It also analyzes significant fires in Rhodes, Attica, Thessaly, Evia, Corfu, and Magnesia, emphasizing the compounded challenges posed by terrain, climate, and human factors in those areas. Additionally, the climate data for each affected area were compared with the weather conditions prevailing at the time of the fires. Copernicus Land Cover and Tree Density data are integrated to aid future management, assessment, and restoration. The analysis of maps and fire statistics underscores a notable pattern: areas with higher pre-fire tree density experienced correspondingly higher burn severity. This research underscores the crucial role of such data in assessing wildfire impact. In addition, compared with Copernicus Emergency Management Service, the burned area maps validate the accuracy and reliability of the utilized satellite data. The total burned area was assessed with a high accuracy rate of 96.28%.

**Keywords:** wildfires; Sentinel-2; Copernicus Land Monitoring Service; Greece; burn severity





## 1. Introduction

From the inception of civilization, humanity has been trying to build and spread its influence across the globe while avoiding the devastating effects of various natural disasters, such as earthquakes, floods, or soil erosion. Wildfires are among the most dangerous natural disasters that threaten humanity and the natural environment, which commonly occur within forested areas [1,2]. In fact, forest fires have often been considered one of the most impactful and prevalent disasters in recent times [3,4], especially as climate change causes a constant increase in temperatures and affects an area's weather conditions. Specifically, climate change causes climatic phenomena, such as droughts and heat waves, which have an impact on the general frequency of wildfire occurrences [5–7]. Additionally, according to Malandra et al. [8], human activities tend to change and shape the natural environment directly due to agricultural [1] or reforestation practices, which affect the abundance of potential natural fuel in the form of tree density. Moreover, unchecked, reckless, and poorly planned urbanization in recent times has also been reported as a main cause of forest fires [3].

Forest fires pose a significant threat to both the ecosystem, the residential humanmade environment, and the local economy [7,9–11]. In particular, aside from the direct danger to

human lives, forest fires are responsible for changes in the behavioral patterns of wildlife, as previous nesting locations may have been lost with the burning of an area [3,12]. Additionally, fires destroy local vegetation and alter soil composition, which may lead to reduced ground intrusion of rainwater, effectively increasing surface runoff and the severity of natural hazards such as the rapid erosion of soil or flooding [5–7,12–16]. Furthermore, patches of burned ground are often irreversibly altered as local flora struggles to regrow, thus slowing down the overall vegetation recovery rate of the affected area [13,14,17,18]. Another significant effect of forest fires on the general environment and local ecosystems is the rapid production of carbon and other pollutants [19]. Specifically, the authors of [2] state that wildfires are an undeniable source of carbon monoxide and dioxide, nitrous oxide, sulfur dioxide, and black carbon emissions, which are extremely detrimental to the ecosystem.

Due to the aforementioned issues, it is imperative to study the effects of forest fires so that proper management of the burnt areas and rapid vegetation recovery can be accomplished by the responsible government bodies [10]. As such, great interest has arisen in the scientific community in researching the severity of forest fire burning. Burn severity refers to the non-immediate effects of a forest fire in an area, which may lead to the loss of soil or organic material aboveground [4,8–10,12,19–21].

This information can be extremely useful for land management and forest fire prevention [10], especially in regions within the Mediterranean, such as Greece, where forest fires have begun breaking out at a more frequent rate in recent years [10] due to the rampant effects of climate change [22]. Some of the most severe and extensive wildfires in Europe have occurred in the Mediterranean areas, according to [8], spanning about 90% of the continent's burned area, increasing the susceptibility to droughts and higher temperatures resulting in frequent and devastating ecological events. Greece's vegetation cover, mainly consisting of brushwood or shrubland and pine forests, in combination with its dry and hot summer temperatures, low humidity and high wind speeds, render it prone to wildfires of significant burn severity [2,7]. As such, Greece experiences numerous wildfires every year, especially during the summer months, resulting in significant destruction, loss of life, and severe damage to the environment. More specifically, during the 2023 fire season, Greece suffered from multiple extremely severe wildfires, nine of the most significant ones occurring in Evros, the island of Rhodes, Dervenochoria, Lagonisi-Kalivia, Loutraki, and Parnitha-Aspropyrgos in Attica, Magnesia, Platanistos of Evia and the island of Corfu.

Burn severity can be measured through field observations; however, this task may be subjective due to human judgment, which can vary from one observer to another, leading to inconsistencies. Additionally, the complex morphology of the affected areas can limit accessibility, particularly in larger districts. Thus, scientific advancements have allowed the use of remote sensing techniques to spatially detect post-fire changes in an area with greater accuracy, increased extent and lower time and financial investment [9,10]. Burn severity can be assessed through a variety of remotely sensed data, such as hyperspectral satellite images provided, often openly, by missions such as Sentinel-2 of the Copernicus service, as demonstrated by several researchers [11,14,23–27]. Most notably, ref. [13] compared the accuracy of fire indices based on Sentinel-2A and Landsat-8 satellite images, concluding that the open data provided by Sentinel-2 proved more reliable in such studies.

One common method for assessing forest fire burn severity is the use of the Normalized Burn Ratio (NBR). This index evaluates pre-fire and post-fire severity in a designated area by analyzing low and high reflectance in the Near Infrared (NIR) and Shortwave-Infrared (SWIR) from satellite images [5,20]. In 2006, ref. [28] proposed using the differenced Normalized Burn Ratio (dNBR) as a potential alternative for identifying changes in post-fire reflectance within a specific area [20]. In 2007, ref. [20] introduced a variation of the previous indices, the Relativized dNBR (RdNBR), designed to more accurately monitor reflectance changes relative to pre-fire conditions [19]. Most recently, Alcaras et al. [26] studied the 2019 summer fires in Sicily, introducing a new index called Normalized Burn Ratio Plus (NBR+). Another significant advancement in the natural disaster management

scientific field is the introduction of Geospatial Intelligence (GEOINT). GEOINT describes the ability to create and present geospatial knowledge by collecting, identifying, and manipulating data for the decision-making environment.

Sometimes, it can be difficult to assess the extent of the damage caused by a forest fire. As such, it can be beneficial for those responsible for managing fires to be informed about the attributes of a burnt area, as it will help prevent or reduce future wildfire threats. Thus, some researchers have opted to compare the burn severity of a forest fire to data such as tree coverage or density and the soil or vegetation type attributed to the burned patches [2,8,23]. One of the most accessible sources of open data on tree density and land cover is provided by the Copernicus Land Monitoring Service, which has been used to research the forest fire of Attica, Greece, in the summer of 2021 [5,29]. The free and openly distributed nature of the Copernicus Land Monitoring data renders them an undeniably useful tool for quick and accurate forest fire burn severity studies.

The aim of the current study is to assess the burn severity of major forest fires of summer 2023 in multiple locations in Greece, including the Attica Region, Loutraki (Gulf of Corinth), Magnesia (Thessaly), Northeast Corfu, Platanistos (Evia), the island of Rhodes and, last but not least, Evros, utilizing open-source data from the Copernicus Land Monitoring Service and the Sentinel-2 satellite mission. Specifically, we implemented the dBNR methodology on two Sentinel-2 satellite images, taken before and after each of the studied forest fires, to assess their burn severity. Additionally, we aspired to contrast the resulting burn severity maps with CORINE Land Cover 2018 (CLC 2018) and Tree Cover Density (TCD) maps, created via data obtained from the Copernicus Land Monitoring Service, in order to assess vegetation types most affected by the flames, as well as the density of the trees in the burned patches. Moreover, a correlation is made between tree cover density data and an affected area's burn severity. We posit that an increase in tree cover density correlates with a corresponding escalation in burning severity. This study seeks to provide an easy and accessible way to increase the information extracted from forest fire burn severity mapping by specifying the necessary land cover and tree cover attributes of the study area using open-source datasets and GEOINT methods.

## 2. Materials and Methods

### 2.1. Study Area

As mentioned in the introduction, this study focuses on the nine most significant wildfire events that occurred in Greece during the 2023 fire season, as presented in Figure 1. The significance of these events is determined by various factors, including the extent of the burned area, the population density, the environmental impact, burn severity, and the region's importance in terms of tourism activities. The wildfires that occurred in northern Greece during the summer of 2023 were the largest ever recorded in the European Union (EU). From the beginning of the year until the present time, these fires have affected approximately 173,000 hectares in Greece [30]. In July specifically, the wildfires in Greece contributed to the emissions of 3.5 million tonnes of $CO_2$ [31], and as a result, these high values were recorded from satellites. Greece's hot and dry climate, combined with strong winds, set favorable conditions for wildfires to occur. Another factor that contributes to wildfires is the apprehension of numerous individuals by the police and fire department due to suspicions of arson [32].

One of the most tragic wildfires in the Greek region occurred during this period in Evros Region (Northeastern Greece), began on 19 August and lasted for more than fifteen days, marking the largest wildfire in the EU since 2000. The smoke plume of the fire reached central Greece and even Crete and southern Italy [33]. On 20 August, the Copernicus Atmosphere Monitoring Service (CAMS) recorded the highest daily emissions compared to the 20-year mean measurements [33]. The wildfire caused extensive damage to 58% of the National Forest Park of Dadia–Lefkimmi–Soufli, and caused 20 casualties [34]. Located in Evros, northeastern Greece, the Dadia–Lefkimi–Soufli Forest National Park covers 428 km$^2$ [35,36] and was established as a protected area in 1980, making it the first protected

forested area in the country [37,38]. Renowned for its biodiversity, it hosts 36 bird of prey species, 166 bird species, and various mammals. It is mostly known for three European species of vultures, the Aegypius monachus, the Neophron percnopterus and the Gyps fulvus, the first of which is one of the last of its kind [35,39,40]. In terms of the vegetation in the Dadia National Park, a wide variety of plants can be found throughout its entire expanse, with a population of 360–400 species and counting [35]. Vegetation includes black pine, Turkish pine, oaks, and maquis shrublands [35]. An estimated 130,000 olive trees (60% of the total 200,000), hundreds of beehives, and thousands of animals, predominantly sheep, goats, and cows, have been burnt. The rare black vulture, once a resident of the affected area, faces an increased threat as the old pine trees crucial for nesting have been lost [41,42]. Dadia–Lefkimi–Soufli Forest National Park, a vital location in Greece for decades, shelters remarkable and endangered species, making it crucial for conservation efforts in Europe. The park, however, had been threatened by two other wildfires recently, occurring on 1 October 2020 and 9 July 2021 in surrounding areas [43]. Hence, Maniatis et al. [43] studied the risk probability of the forest reserve being affected by wildfires, and they classified the park's area as being at high risk due to wildfires in multiple locations.

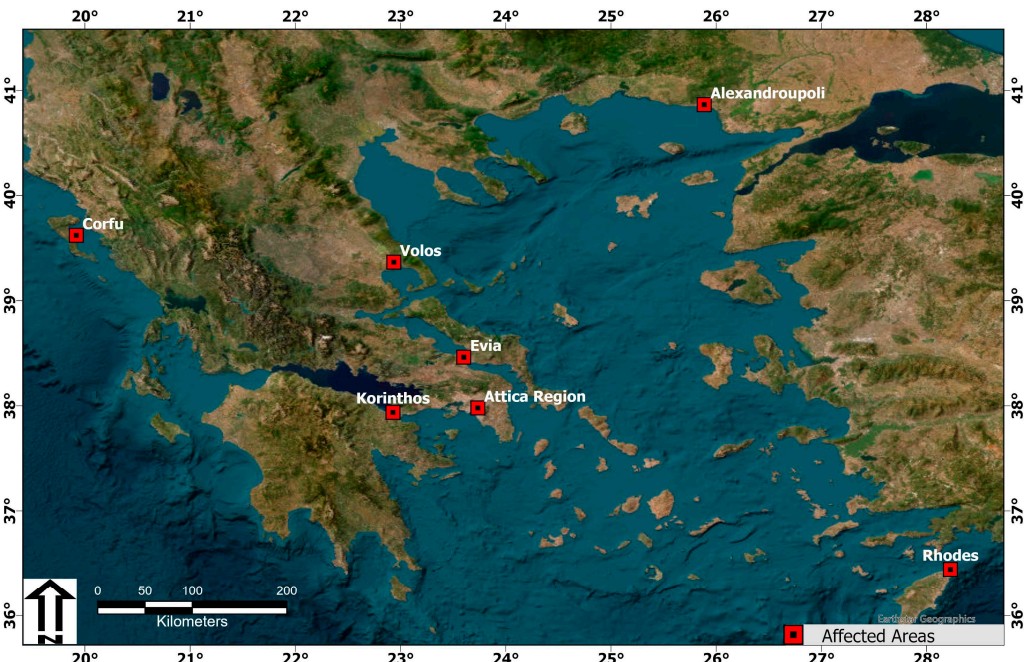

**Figure 1.** Affected Areas from the nine most significant wildfire events in Greece during 2023 (source image: Google Earth).

In July and August, wildfires wreaked havoc in various Greek regions. A ten-day wildfire on Rhodes, starting on July 18, prompted widespread evacuations as flames, fueled by high temperatures and local winds, spread across the island, affecting the southeastern coastal areas. Thousands of tourists fled, with 16,000 rescued by land and 3000 by sea [44,45]. This fire resulted in the destruction of around 50,000 olive trees and the loss of 2500 animals and beehives [46]. In Attica, three fires erupted on 17 July in Dervenochoria, Lagonisi-Kalivia, and Loutraki, followed by a destructive wildfire on 22 August in Fyli, Central Attica. This fire, extending to Parnitha and Aspropyrgos, damaged houses, forests, and the National Park of Parnitha [47]. This wildfire inflicted significant damage to houses, surrounding forested areas, and the frequently impacted National Park of Parnitha in the past [48]. Although no human casualties occurred, the burned areas significantly impacted the sustainability of Attica, home to 3,792,469 people. In the Lagonisi wildfire, over a dozen cats and dogs perished at a privately run shelter for strays. The impact of the Parnitha wildfire was extensive, affecting over 1000 species of plants and animals, including red deer and wolves, along with various reptiles, birds, and mammals recorded in the region [42,49].

Another extensive wildfire struck the Magnesia prefecture in Thessaly on 26 July, causing extensive damage and reaching army facilities. In the Magnesia Wildfire, the tragic loss of life exceeded 3000 animals, predominantly sheep and goats, in Agios Georgios Feron. The devastating impact resulted in an 80% loss of Magnesia's livestock and the burning of over 50,000 acres, including arable land, agricultural areas, and forested lands [50]. Tragically, on 23 July, a wildfire in Southeastern Evia resulted in a Canadair aircraft crash, claiming two lives, evacuating Platanistos, and threatening other residential areas. Notably, wildfires affected a large part of Northern Evia two years ago [45]. The same day, a wildfire was witnessed on the northeastern side of Corfu near the Peritheia region, causing damage and evacuating approximately 2500 people [45,51].

It is necessary to note that the weather conditions during each fire incident were compared to the average climate data for each affected area. The climate data utilized were derived from weather reports 1985–2015 for Rhodes, Corfu, and Evros and 2005–2015 for the remaining areas. In all cases, the actual maximum daily temperature exceeded the average maximum, along with higher wind speeds. In addition, the humidity on the days of the fires was higher than average in Corfu, Platanistos, Magnesia, and Evros and below average in the other areas [52–54].

*2.2. Data Used*

This study utilizes open-access data, including satellite images from Copernicus' Sentinel-2 and land information data. These datasets were processed at various stages during the implementation of the methodology. The specific data used for this study are detailed in Table 1. All the datasets employed in this study are freely available and easily accessible to anyone interested in replicating the methodological steps outlined below.

**Table 1.** Employed Data.

| Data | Format | Resolution | Source |
| --- | --- | --- | --- |
| Sentinel-2 Imagery | Optical Level-2A | 10 m | Copernicus Open Access Hub |
| CORINE Land Cover (CLC 2018) | Vector (Polygon) | - | Copernicus Land Monitoring Service |
| Tree Density | Raster | 10 m | Copernicus Land Monitoring Service |

The mapping of the wildfires was accomplished using optical/multispectral Sentinel-2 mission images of the ESA Copernicus program. These images used to be accessible from the platform of Copernicus Open Access Hub [55]. All the datasets of the Sentinel missions of the ESA Copernicus program are now accessible from the Dataspace Copernicus [56]. The Sentinel-2A products were atmospherically corrected, and the scene was classified as Level-2A Bottom-of-Atmosphere (BOA) [57].

The two most important and positive aspects of the Sentinel mission and the images it provides, which greatly influenced the decision to use Sentinel-2 data for the wildfires, are the spatial resolution of the images and the temporal repeatability. Sentinel-2 images have a spatial resolution of 10 m in some of the bands, and every 5 days, there is a satellite image of the area of interest. The images from Sentinel-2A utilized for all the wildfires are presented in Table 2. The images were selected depending on the cloud cover. The aim was that the cloud cover of the images was as low as possible and that no clouds obscured the study areas.

CORINE Land Cover data, which contained the information on the land cover of all the areas of interest, were acquired from the geodatabase of the Copernicus Land Monitoring Service. Copernicus Land Monitoring Service is an open-access geodatabase, and all the data are free for the users. The CLC 2018 was in vector polygon form with very high accuracy (>= 85%) and a minimum mapping unit of 25 hectares (0.25 km$^2$) [58]. In total,

CORINE Land Cover has 44 land cover classes and a minimum mapping width of 100 m. CLC 2018 was used to depict the affected areas of the wildfire and what kind of land cover was burned.

Tree Cover Density with a spatial resolution of 10 m was utilized to map the forested areas and to understand the density of the forests burnt by the wildfires. TCD (Tree Cover Density) is a raster data file that provides information on the proportional tree coverage per pixel at 10 m [59]. The coverage range starts from 0% (all non-tree-covered areas) to 100% (areas fully covered by trees).

In this study, the processing of the satellite images was completed via ESA STEP SNAP v9.0 software. SNAP software is open and free for all users, provided by the European Space Agency (ESA). Additionally, for the mapping of the results, the commercial software of ESRI, ArcGIS Desktop v.10, was utilized.

**Table 2.** Sentinel-2 Images for Each Area.

| Area | Wildfire Start Date | Sentinel-2 Acquisition Date |
|---|---|---|
| Corfu | 23 July 2023 | 20 July 2023 30 July 2023 |
| Dervenochoria | 17 July 2023 | 14 July 2023 24 July 2023 |
| Evros | 19 August 2023 | 29 July 2023 (T35TLF, T35TMF) 12 September 2023 (T35TLF, T35TMF) |
| Lagonisi | 17 July 2023 | 4 July 2023 19 July 2023 |
| Loutraki | 17 July 2023 | 2 July 2023 27 July 2023 |
| Magnesia | 26 July 2023 | 22 July 2023 21 August 2023 |
| Parnitha | 22 August 2023 | 18 August 2023 28 August 2023 |
| Platanistos (Evia) | 23 July 2023 | 19 July 2023 29 July 2023 |
| Rhodes | 18 July 2023 | 8 July 2023 (T35SNA, T35SNV) 2 August 2023 (T35SNA, T35SNV) |

*2.3. Methodology*

A commonly employed approach for gauging the severity of a forest fire involves utilizing the Normalized Burn Ratio (NBR). This index assesses the pre-fire and post-fire severity of a designated area by analyzing the low and high reflectance of objects in the Near Infrared (NIR) and Shortwave-Infrared (SWIR), respectively, as captured in satellite imagery [5,20].

In 2006, ref. [17] examined the potential of NBR as a spectral index for assessing burn severity, challenging its accuracy in relation to vegetation displacement post-fire. In the same year, ref. [28] suggested using the differenced Normalized Burn Ratio (dNBR) as a potential alternative for detecting post-fire reflectance alterations in an area [19]. In 2007, ref. [20] introduced a new iteration of the previous indices, termed Relativized dNBR (RdNBR), with the aim of more precisely monitoring reflectance changes in relation to the conditions existing before the forest fire [19]. The RdNBR index was further developed by [60] as a newer approach [19].

In the summer of 2019, Alcaras et al. [26] conducted a study on wildfires in Sicily, Italy. During this investigation, they unveiled a novel index known as Normalized Burn Ratio Plus (NBR+). Despite NBR's shortcomings, it remains a broadly utilized index for assessing wildfire burn severity, either independently [21,23] or in combination with other indices [10,14], making it an irreplaceable tool for post-fire reflectance changes.

GEOINT, as a fundamental principle, involves manipulating and combining all available data, including geospatial and satellite imagery, to create useful products for planning, decision-making, and emergency response [61]. Remote sensing capabilities in wildfires, particularly for mapping burned areas, are widely employed [5,62]. Additionally, GEOINT allows the use of multispectral data and spectral indices, such as the burned area index (BAI) [5] and the aforementioned NBR, dNBR, or RdNBR, to retrieve further information through geospatial analysis.

The following schematic flowchart (Figure 2) outlines the methodology that was implemented for the analysis of the burned areas, utilizing free satellite data and combining SNAP and ArcGIS Desktop software. According to the a, b, and c steps of the diagram, the preprocessing step was applied in order to create cloud masks for the pre- and post-fire satellite imagery. Following this, the images were resampled (step d) to a resolution of 10 m and the area of interest (AOI) was defined by coordinates while applying the cloud masks (step e).

First of all, it was essential to create the cloud masks for the images in order to not have misleading results. After that, the resampling at 10 m was conducted with the nearest neighbor method and the subset of the images, leaving only the Areas of Interest and the Bands, which were necessary for image processing. The calculation of the Normalize Burn Ratio (NBR) was the last step of preprocessing, and the calculations were conducted with the Band Math tools (step f). NBR is computed with the spectral bands of near-infrared (NIR) and shortwave-infrared (SWIR). In the NIR, the burned areas have low reflectance, but on the other hand, the SWIR has high reflectance. The mathematical Equation (1) shows the healthy vegetation with high values in the area of interest. Low values in burned areas mean low or no vegetation [5,26,63–65].

$$NBR = \frac{NIR - SWIR}{NIR + SWIR} \tag{1}$$

From the *NBR* of the pre-fire and post-fire images, the difference in *NBR* (step j) can be calculated (*dNBR*). Equation (2) provides the burn severity of the wildfire. Burn severity is the degree of a wildfire's impact on an area's ecosystem. It is crucial to estimate the Burn Severity since it provides information that is useful for forest restoration attempts and natural disaster management [5,66–69]. The categories of Burn Severity according to *dNBR* values are presented in Table 3

$$dNBR = Pre fire NBR - Post fire NBR \tag{2}$$

Lastly, the relativized burn ratio (*RBR*) was estimated (step k) according to Equation (3). The *RBR* aims to enhance the accuracy of the burn severity and helps to distinguish the changes after a fire in low vegetation regions. The results of *RBR* were masked using Equation (3) in order to remove any possible cloud in the area of interest, which could lead to a distortion of the final results [5,60,70].

$$RBR = \frac{dNBR}{pre fire NBR + 1.001} \tag{3}$$

Furthermore, the mask can take away the areas that contain water bodies that can be attributed falsely to burned areas. The following step was to import the images in ArcMap software for the reclassification (steps n,o) of the results according to Table 3. When the classification was conducted, it was necessary to convert raster data to a vector in order to estimate the burn areas and extract the final burned area (step q). Using the vector data of

the final burned area, we managed to clip the CLC 2018 of the areas and the Tree Cover Density to estimate the total damage.

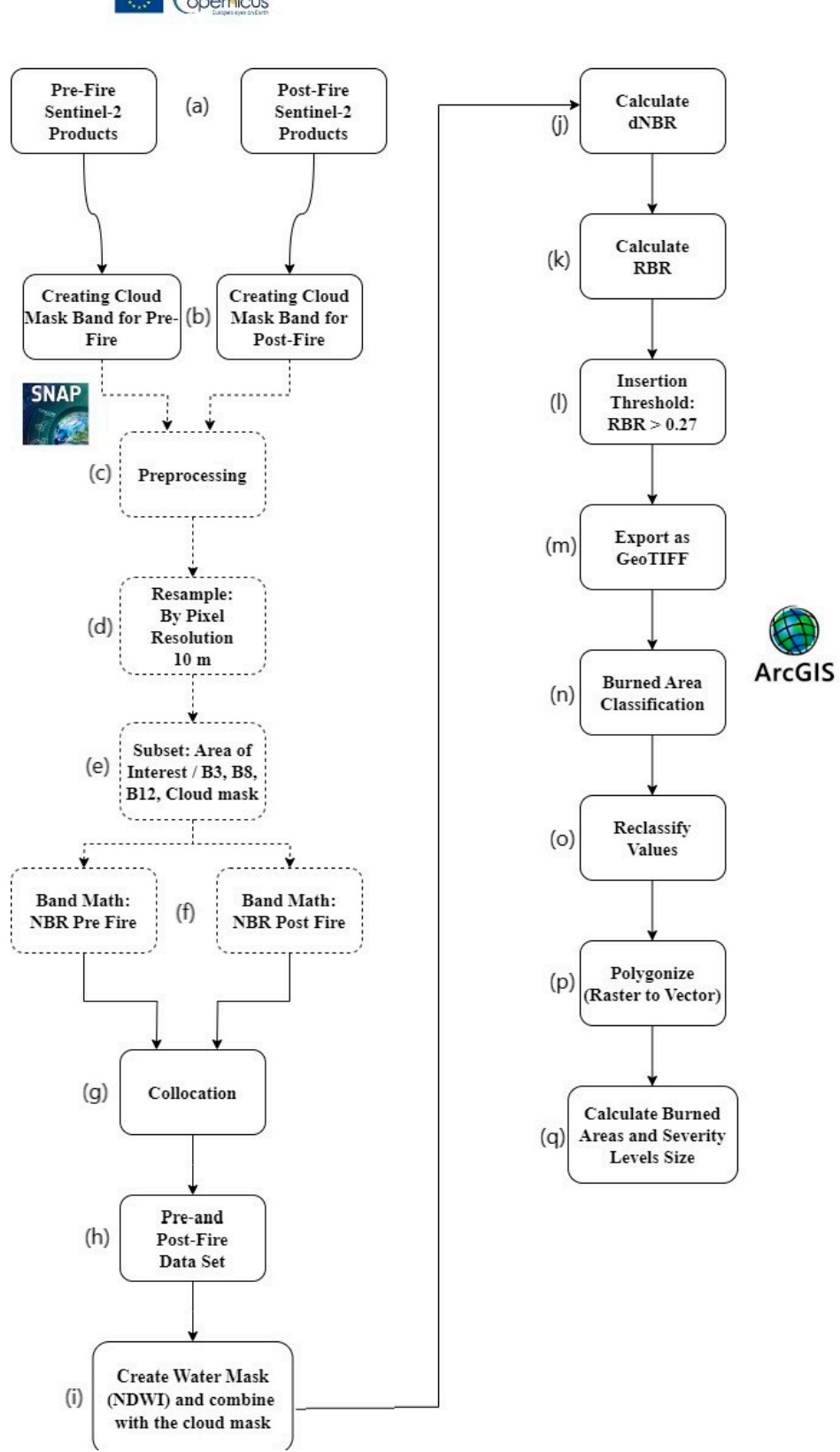

**Figure 2.** Methodology Flowchart.

**Table 3.** Burn Severity classification according to dNBR values [71].

| dNBR Value | Burn Severity |
|---|---|
| 0.099 | Unburned Areas |
| 0.100–0.269 | Low Severity |
| 0.27–0.439 | Moderate–Low Severity |
| 0.440–0.659 | Moderate–High Severity |
| 0.660–1.300 | High Severity |

## 3. Results

### 3.1. Burned Areas and Burn Severity

The total burned area (Table 4) of the nine wildfires in Greece that occurred and have been examined during the 2023 fire season reached 1.415,63 km$^2$, making the process of planning, preventing, and fighting fires essential due to the large extent of damage caused to the region. Fire has, until recently, been viewed primarily as a destructive force that should, for the most part, be suppressed or excluded from the forest. The burn severity of each affected area is presented in the following tables (Tables 5–9). As Figures 3 and 4 present, the wildfire in Evros had the most devastating impact, reaching 903.04 km$^2$ of burned area, followed by Rhodes (Figures 5 and 6) with 172.77 km$^2$, Dervenochoria (Figures 7 and 8) with 111.53 km$^2$, Magnesia (Figures 9 and 10) with 82.40 km$^2$, Parnitha (Figures 11 and 12) with 56.97 km$^2$, Lagonisi (Figures 11 and 12) with 36.47 km$^2$, Northeast Corfu (Figures 9 and 10) with 21.71 km$^2$, Platanistos (Figures 5 and 6) with 19.78 km$^2$ and Loutraki (Figures 7 and 8) with 10.93 km$^2$.

**Table 4.** The main wildfires in Greece in Summer 2023.

| Start Date | Location | Burned Area (km$^2$) | Most Burned Land Type | Validation–Burned Area (km$^2$) |
|---|---|---|---|---|
| 17 July | Dervenochoria, West Attica | 111.53 | Sclerophyllous Vegetation | 117.09 |
| 17 July | Lagonisi–Kalivia, East Attica | 36.47 | Herbaceous Vegetation Associations | 38.69 |
| 17 July | Loutraki, Gulf of Corinth | 10.94 | Transitional Woodland–Shrub | 11.96 |
| 18 July | Rhodes, Dodecanese | 172.77 | Transitional Woodland–Shrub | 177.74 |
| 23 July | Northeast Corfu | 21.71 | Sparsely Vegetated Areas | 21.77 |
| 23 July | Platanistos, Evia | 19.79 | Sclerophyllous Vegetation | 19.68 |
| 26 July | Magnesia, Thessaly | 82.41 | Sclerophyllous Vegetation | 82.64 |
| 19 August | Dadia, Evros | 903.04 | Mixed Forest | 938.81 |
| 22 August | Parnitha–Aspropyrgos, Central Attica | 56.98 | Transitional Woodland–Shrub | 61.93 |
| **Total Burned Area:** | | 1415.63 | | 1470.31 |

**Table 5.** Dadia–Evros Burn Severity Statistics.

| | Evros | |
|---|---|---|
| **Burn Severity** | **Area (km$^2$)** | **Percentage (%)** |
| Low | 234.32 | 25.95 |
| Moderate–Low | 206.61 | 22.88 |
| Moderate–High | 337.40 | 37.36 |
| High | 124.71 | 13.81 |

**Table 6.** Loutraki–Dervenochoria Burn Severity Statistics.

| Loutraki | | | Dervenochoria | | |
|---|---|---|---|---|---|
| **Burn Severity** | **Area (km²)** | **Percentage (%)** | **Burn Severity** | **Area (km²)** | **Percentage (%)** |
| Low | 3.04 | 27.77 | Low | 23.58 | 21.14 |
| Moderate–Low | 3.5 | 32.02 | Moderate–Low | 55.72 | 49.96 |
| Moderate–High | 4.4 | 40.22 | Moderate–High | 32.04 | 28.73 |
| High | 0 | 0 | High | 0.193 | 0.17 |

**Table 7.** Parnitha–Lagonisi Burn Severity Statistics.

| Parnitha | | | Lagonisi | | |
|---|---|---|---|---|---|
| **Burn Severity** | **Area (km²)** | **Percentage (%)** | **Burn Severity** | **Area (km²)** | **Percentage (%)** |
| Low | 20.46 | 35.90 | Low | 13.78 | 33.78 |
| Moderate–Low | 33.60 | 58.96 | Moderate–Low | 16.87 | 46.25 |
| Moderate–High | 2.92 | 5.13 | Moderate–High | 5.83 | 15.98 |
| High | 0.00 | 0.00 | High | 0.00 | 0.00 |

**Table 8.** Rhodes–Evia Burn Severity Statistics.

| Rhodes | | | Evia | | |
|---|---|---|---|---|---|
| **Burn Severity** | **Area (km²)** | **Percentage (%)** | **Burn Severity** | **Area (km²)** | **Percentage (%)** |
| Low | 34.15 | 19.77 | Low | 2.11 | 10.66 |
| Moderate–Low | 54.13 | 31.33 | Moderate–Low | 4.77 | 24.13 |
| Moderate–High | 82.08 | 47.51 | Moderate–High | 11.91 | 60.19 |
| High | 2.41 | 1.39 | High | 1 | 5.03 |

**Table 9.** Corfu–Magnesia Burn Severity Statistics.

| Corfu | | | Magnesia | | |
|---|---|---|---|---|---|
| **Burn Severity** | **Area (km²)** | **Percentage (%)** | **Burn Severity** | **Area (km²)** | **Percentage (%)** |
| Low | 11.63 | 53.55 | Low | 48.73 | 59.14 |
| Moderate–Low | 8.40 | 38.69 | Moderate–Low | 32.45 | 39.38 |
| Moderate–High | 1.68 | 7.76 | Moderate–High | 1.22 | 1.48 |
| High | 0.00 | 0.00 | High | 0.00 | 0.00 |

Analyzing the burn severity of these wildfires, according to all the figures of burned area and burn severity, the impact was noticeable with a moderate–high severity accounting for 33.87% or 479.48 km² of the total burned area, moderate–low for 29.39% or 416.05 km², low for 27.68% or 391.79 km² while high severity was 9.06% or 128.31 km². In particular, the moderate–high severity seems to be dominant, while the burned area seems to be equally distributed among the moderate–high, moderate–low, and low severity levels, though the high severity levels must be highlighted.

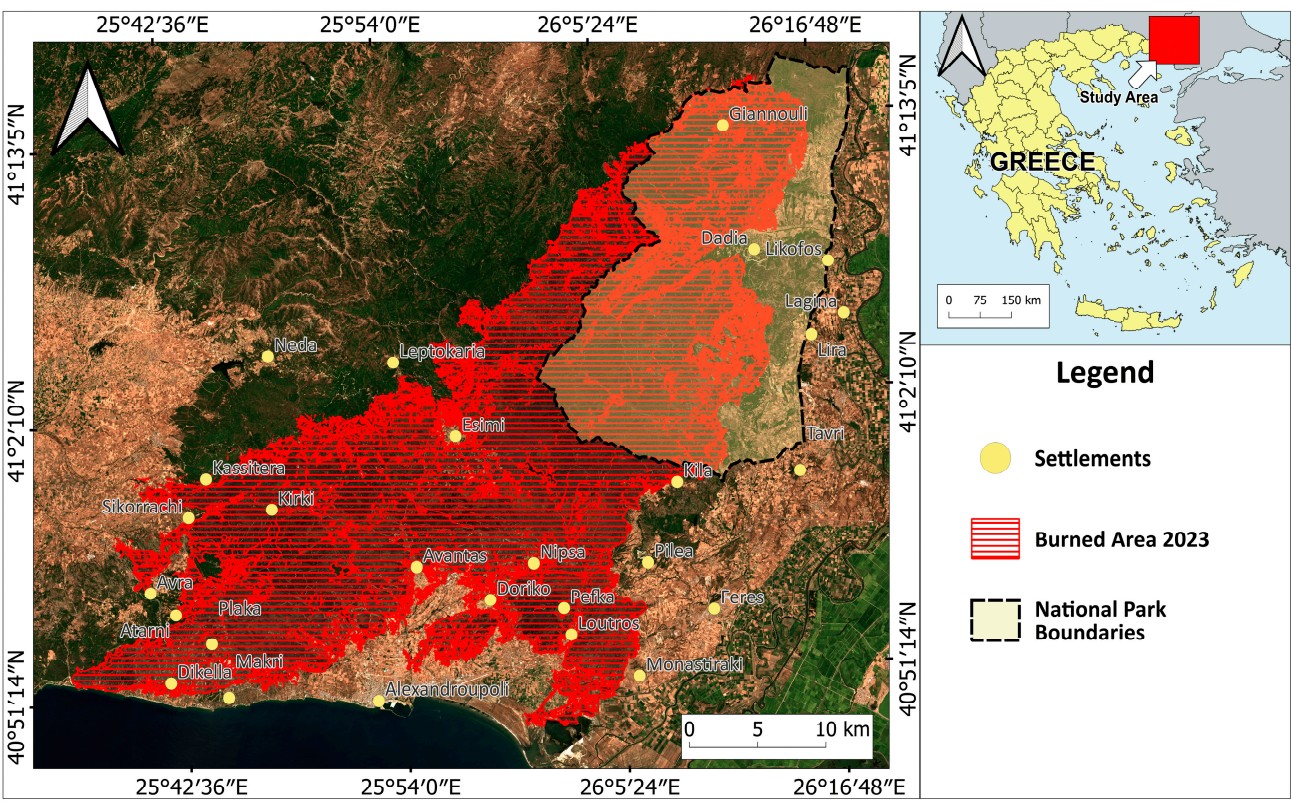

**Figure 3.** Dadia–Evros Burned Area.

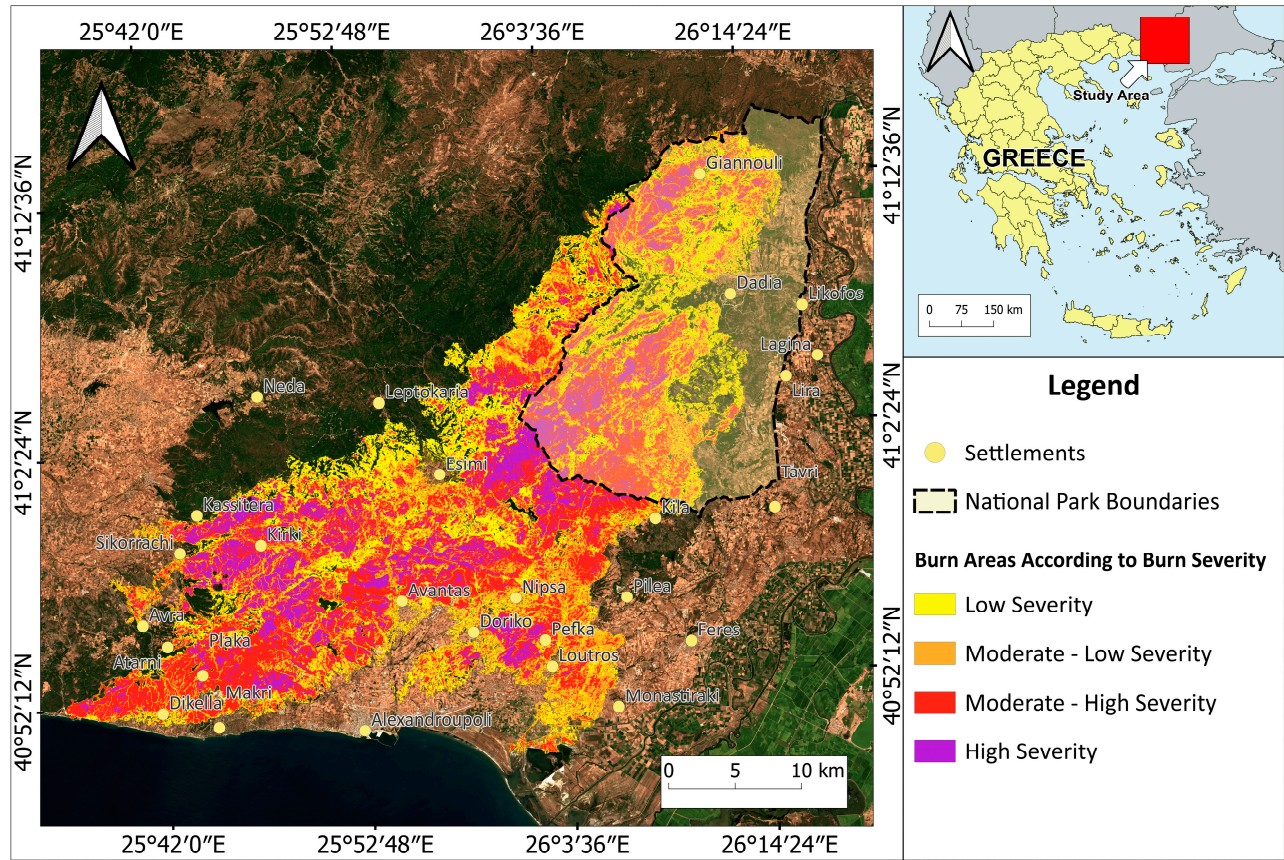

**Figure 4.** Dadia–Evros Burn Severity of the Wildfire.

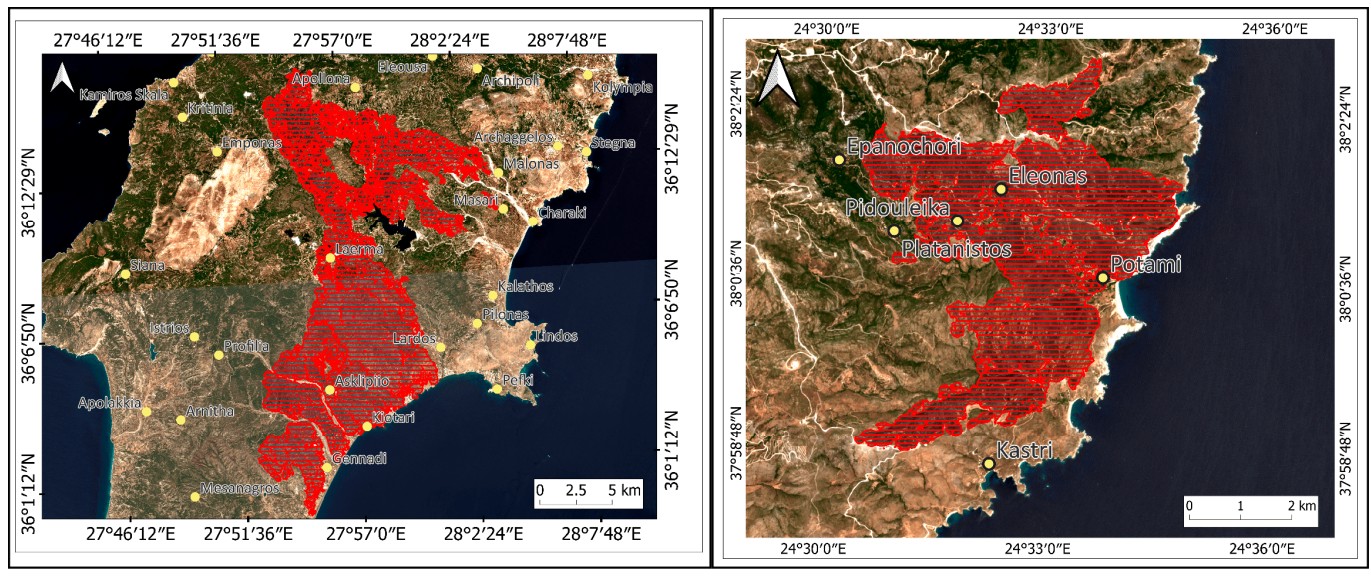

**Figure 5.** Rhodes–Evia Burned Area.

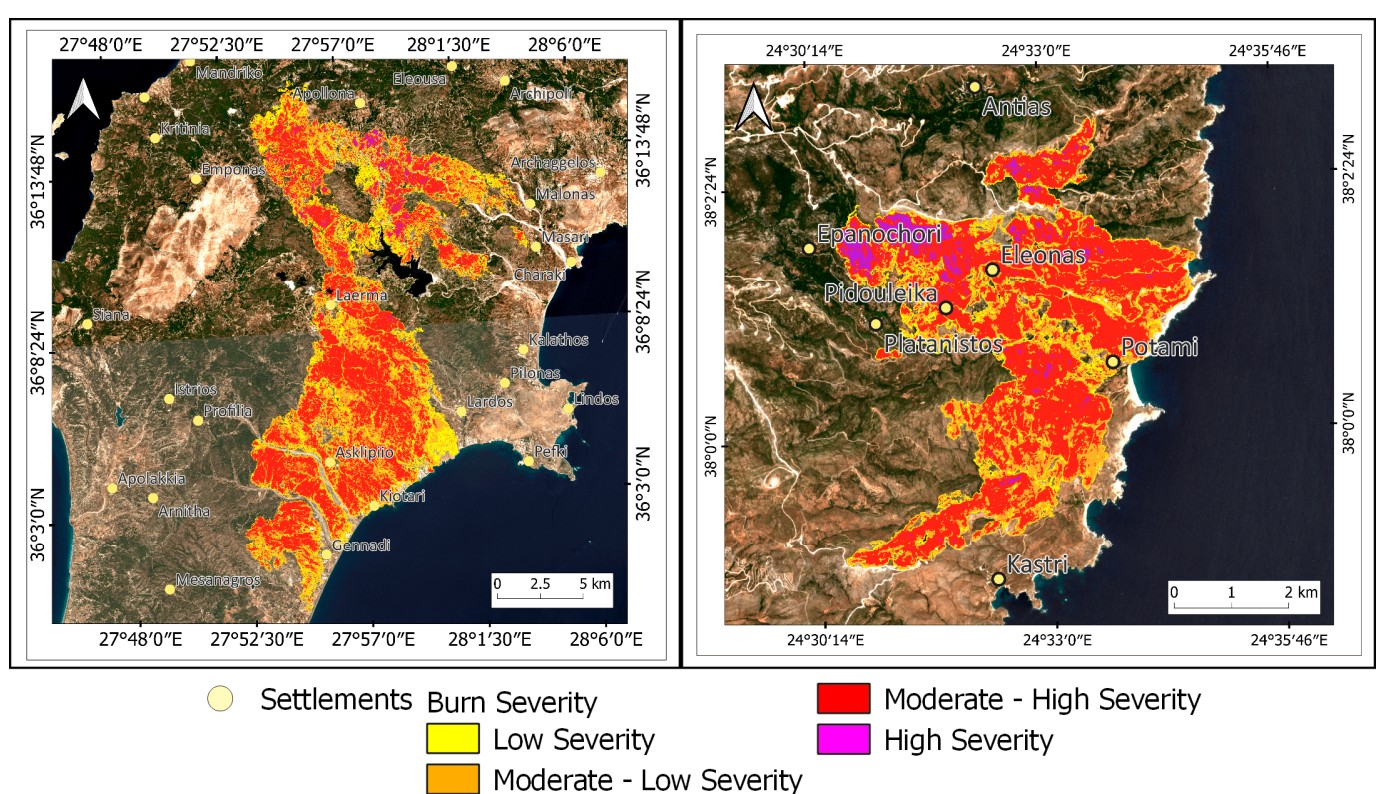

**Figure 6.** Rhodes–Evia Burn Severity.

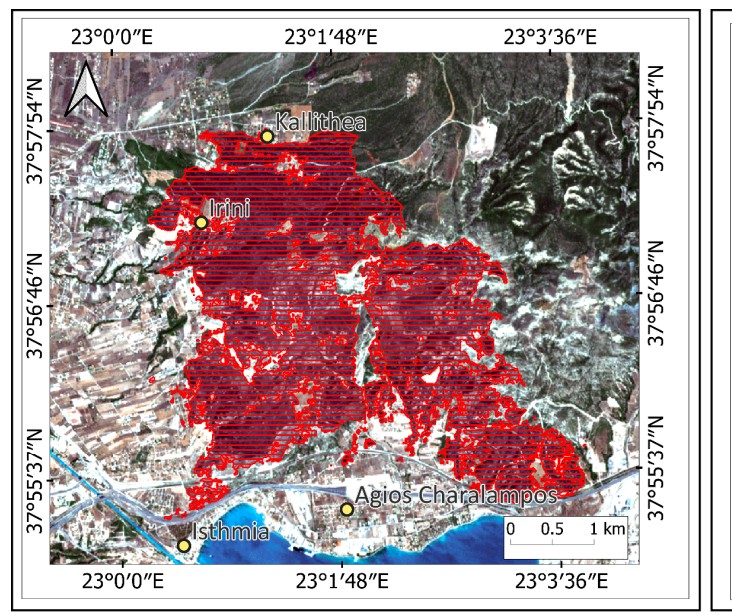
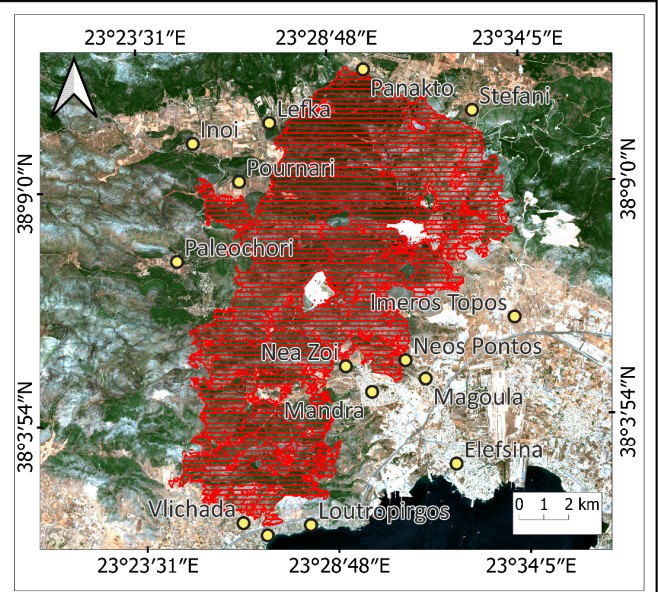

○ Settlements ≡ Burned Area

**Figure 7.** Loutraki–Dervenochoria Burned Area.

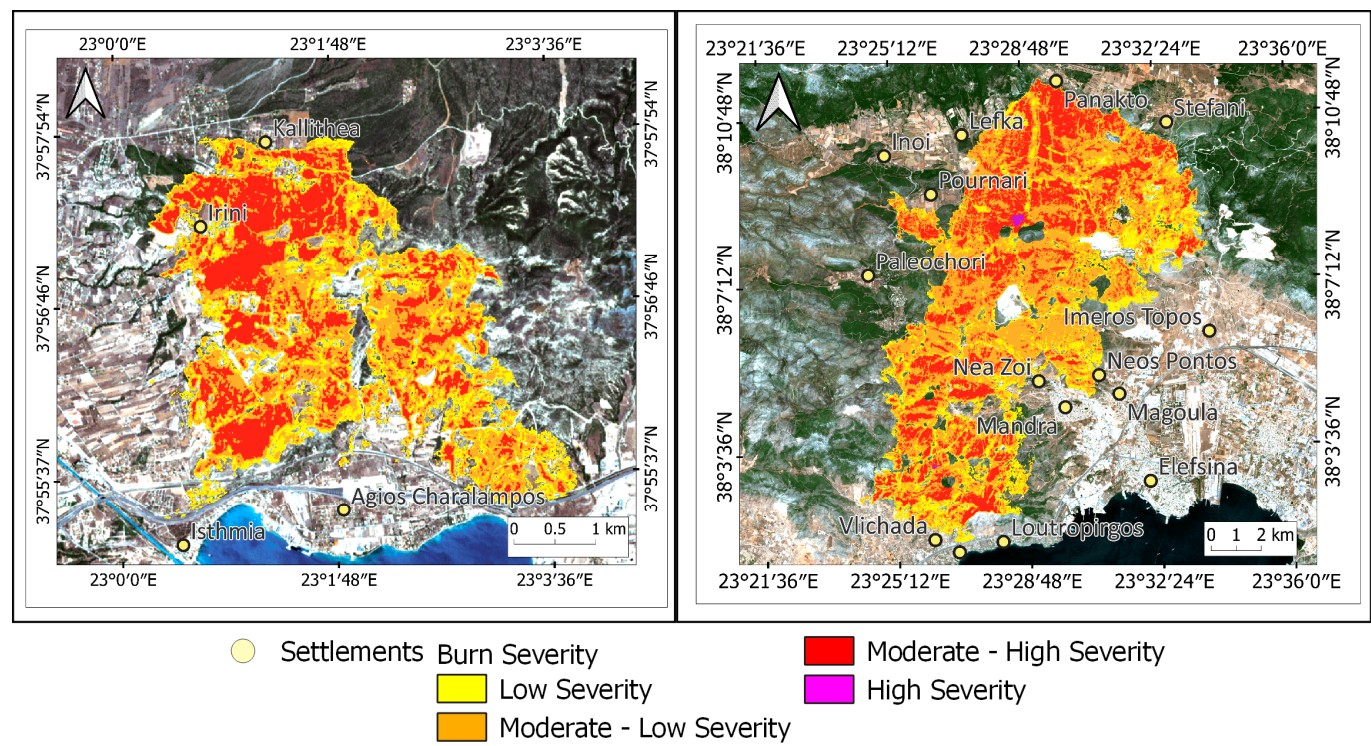

○ Settlements Burn Severity ▮ Moderate - High Severity
▮ Low Severity ▮ High Severity
▮ Moderate - Low Severity

**Figure 8.** Loutraki–Dervenochoria Burn Severity.

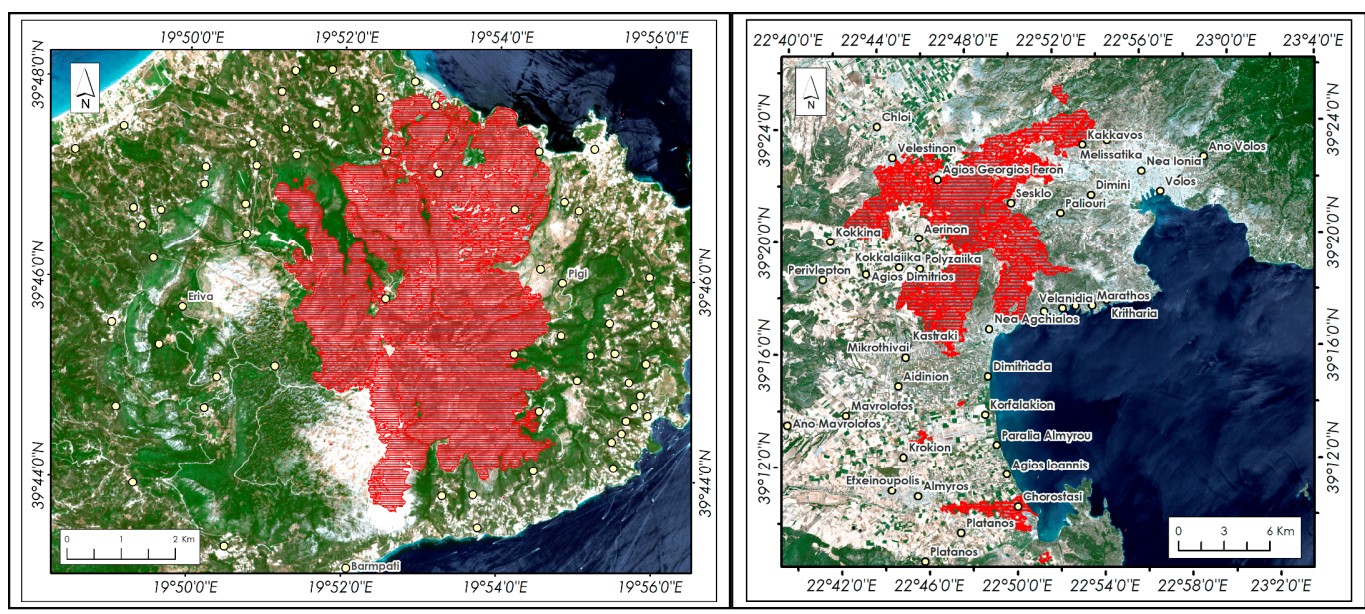

**Figure 9.** Corfu–Magnesia Burned Area.

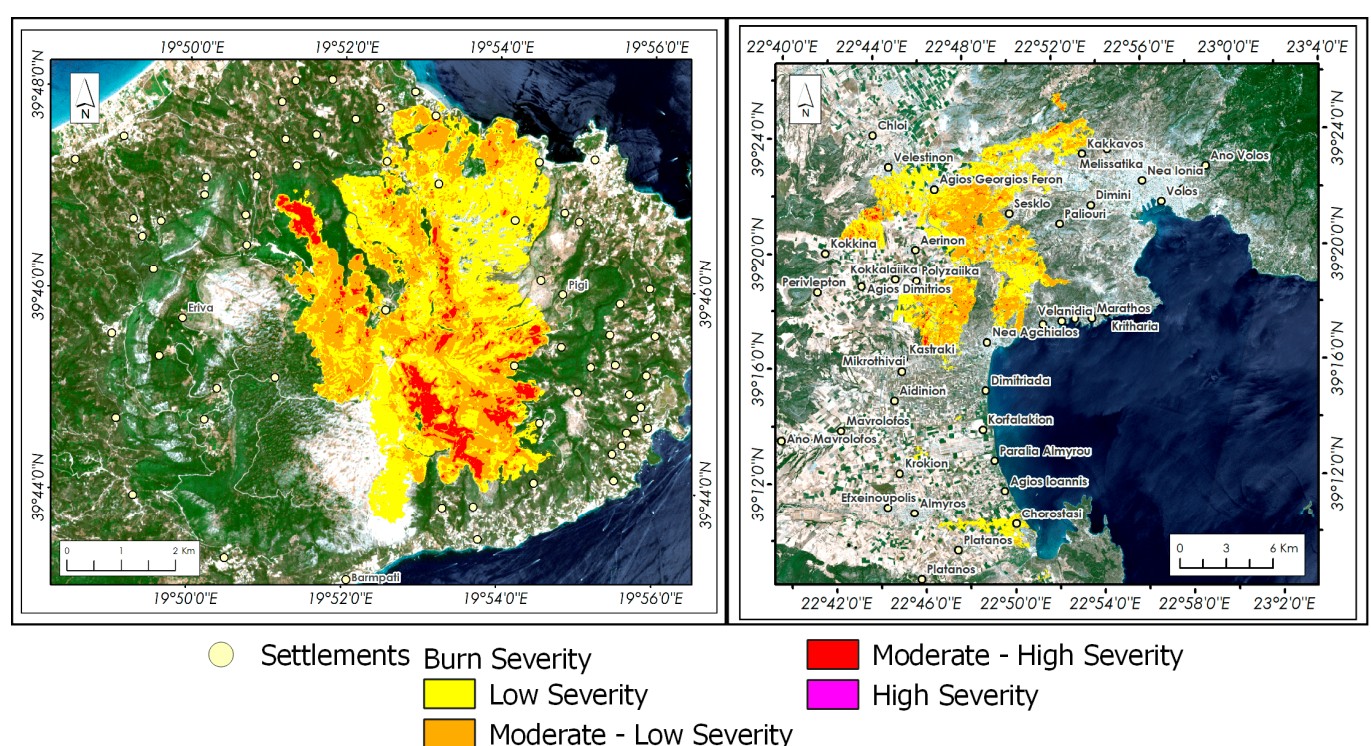

**Figure 10.** Corfu–Magnesia Burn Severity.

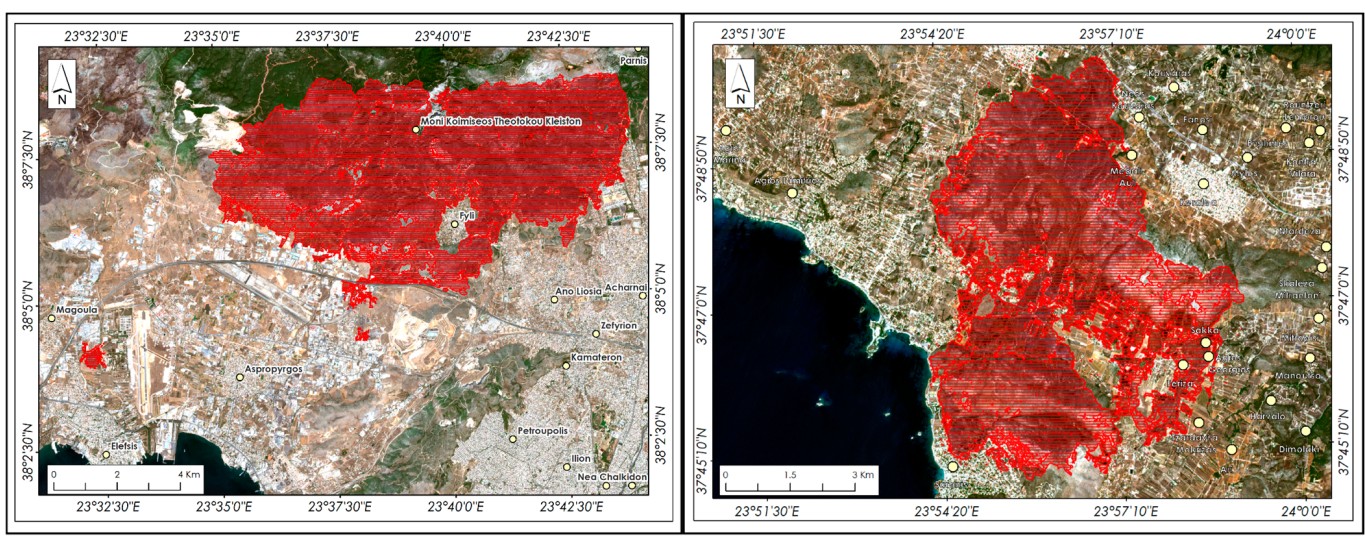

**Figure 11.** Parnitha–Lagonisi Burned Area.

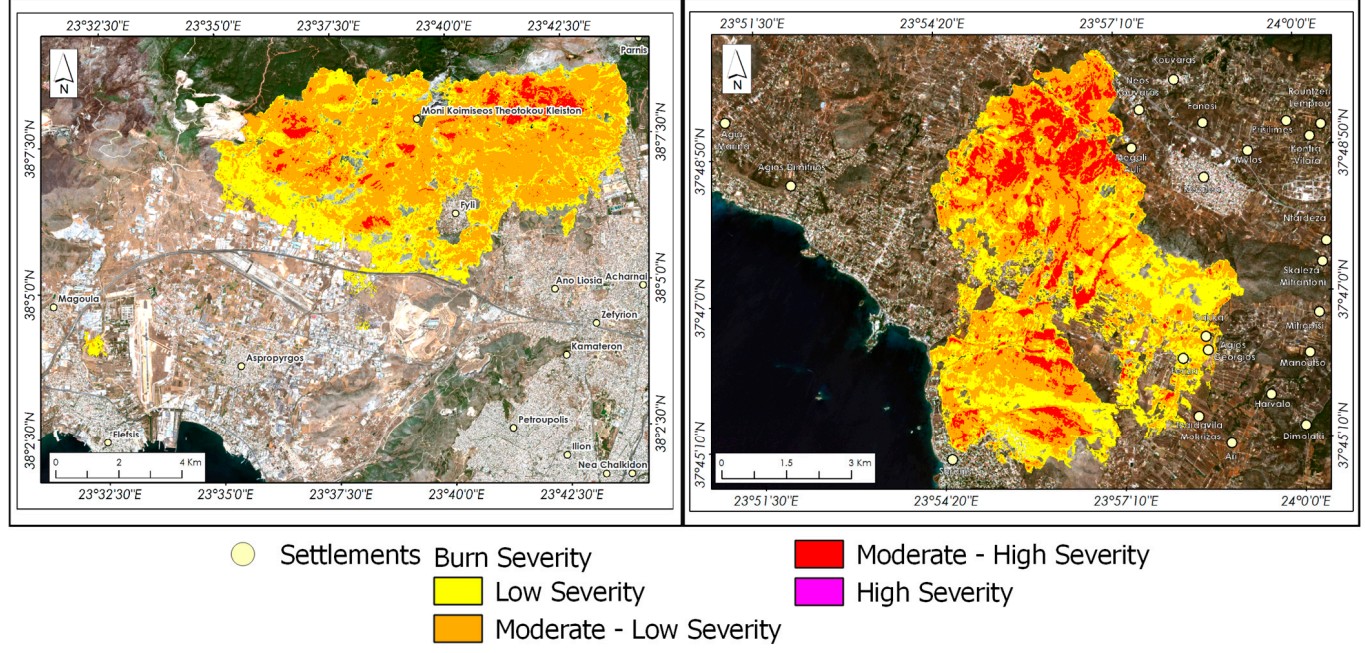

**Figure 12.** Parnitha–Lagonisi Burn Severity.

### 3.2. Land Cover of the Burned Area

Regarding the affected land cover based on CORINE Land Cover 2018 as presented in the figures, the total numbers show that the third CLC category, which includes broad-leaved forests, coniferous forests, mixed forests, natural grasslands, moors and heathlands, Sclerophyllous vegetation, traditional woodland–shrub, and sparsely vegetated areas is heavily impacted by the wildfires covering 81.38% or 1,152,07 km$^2$ of the total burned area. Figure 13 presents the categories of CORINE Land Cover Legend. More specifically, the most affected category in the wildfire of Corfu (Figure 14) was the Sparsely vegetated areas with 13.11 km$^2$ or 60.39%, while in Dervenochoria (Figure 15) was Sclerophyllous vegetation with 36.80 km$^2$ or 33.01%, in Evros (Figure 16) was Mixed forest with 244.96 km$^2$ or 27.12%, in Lagonisi (Figure 17) was herbaceous vegetation associations with 29.06 km$^2$

or 79.70%, in Loutraki (Figure 15) was Transitional woodland–shrub with 5.70 km² or 52.16%, in Magnesia (Figure 14) was Sclerophyllous vegetation with 29.14 km² or 35.37%, in Parnitha (Figure 17) was Transitional woodland–shrub with 19.54 km² or 34.30%, in Platanistos (Figure 18) was Sclerophyllous vegetation with 10.95 km² or 55.33% and in Rhodes (Figure 18) was a transitional woodland–shrub with 76.52 km² or 44.29%. The land cover of each study area is presented in the following tables (Tables 10–14).

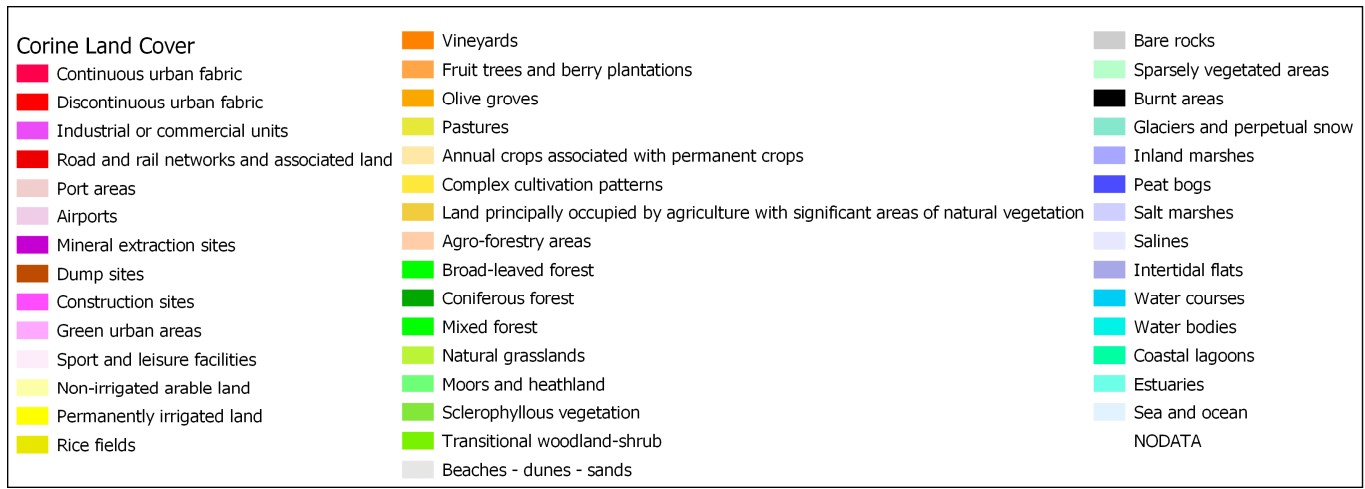

**Figure 13.** CORINE Land Cover Legend.

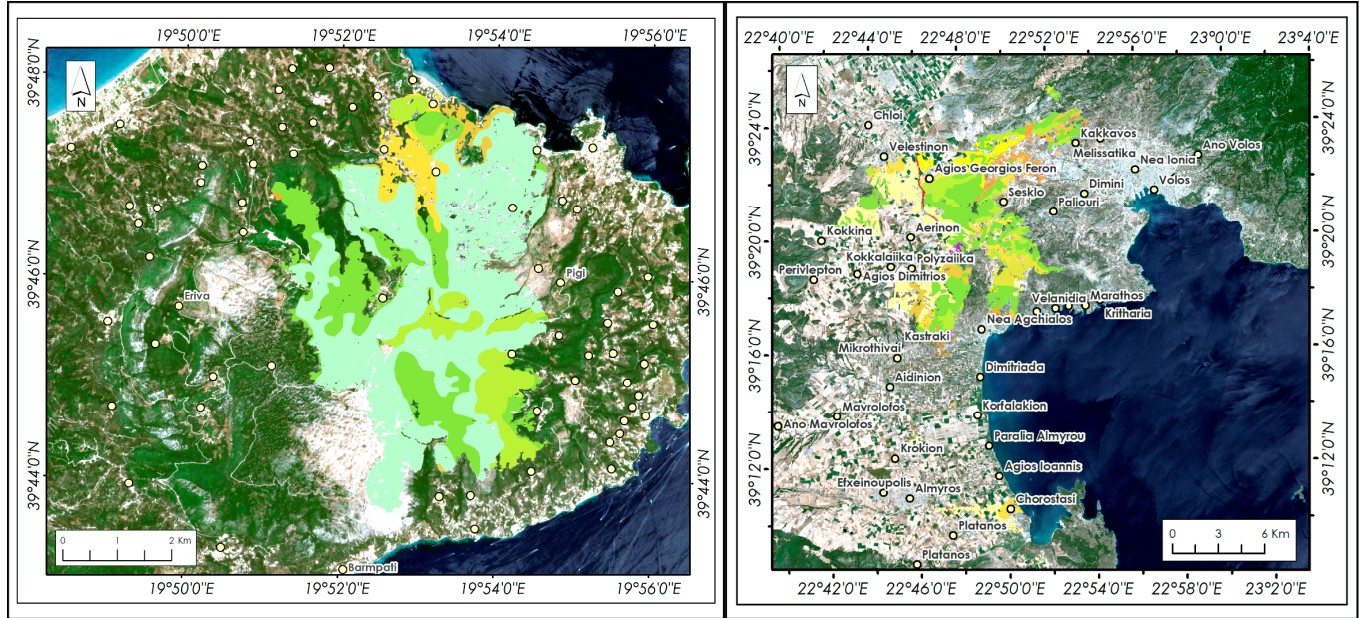

**Figure 14.** Corfu–Magnesia CORINE Land Cover of the Burned Area.

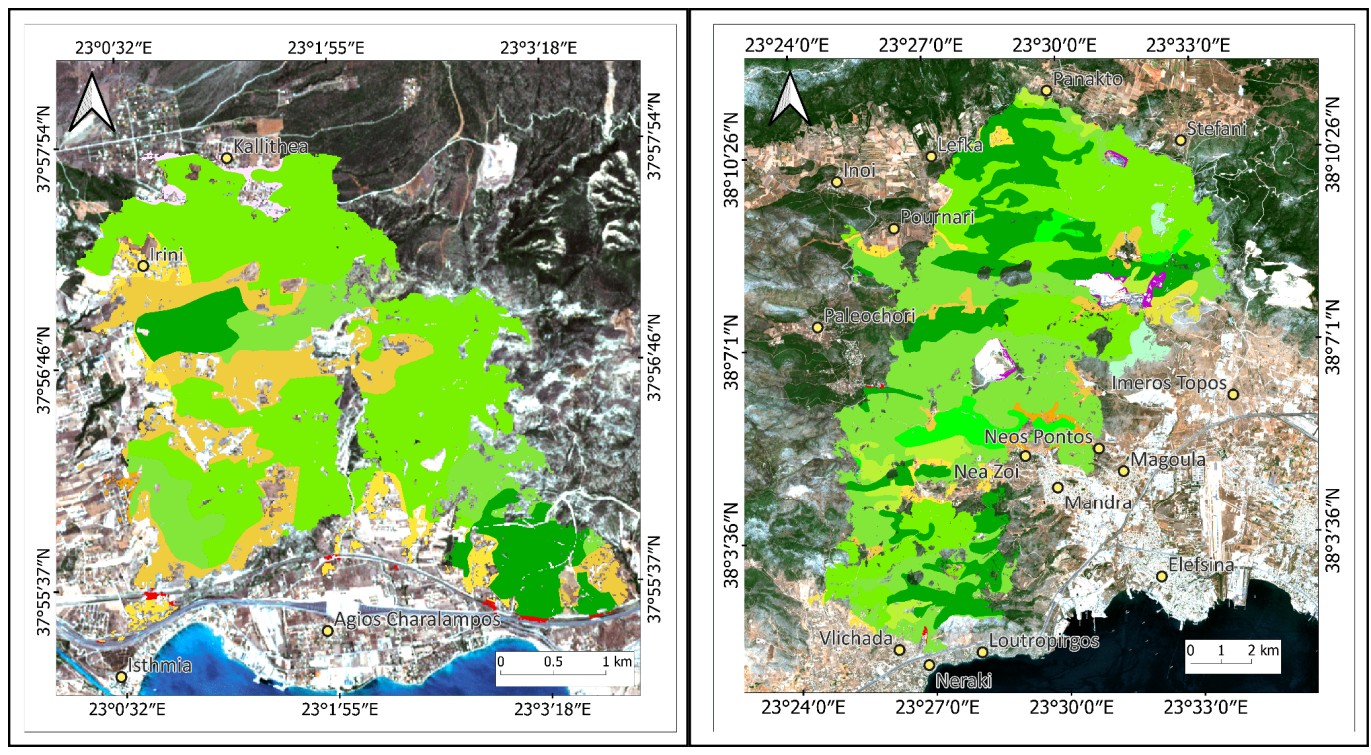

**Figure 15.** Loutraki–Dervenochoria CORINE Land Cover of the Burned Area.

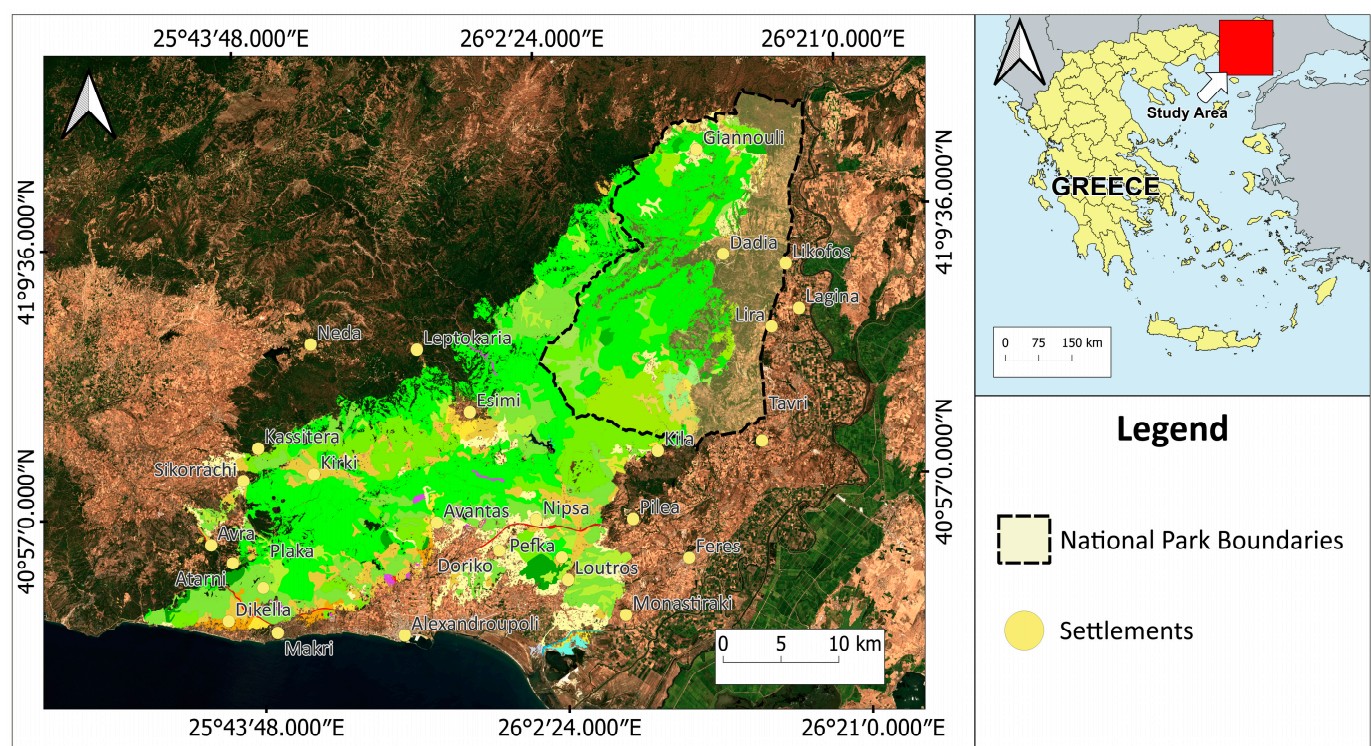

**Figure 16.** Dadia–Evros Corine Land Cover of the Burned Area.

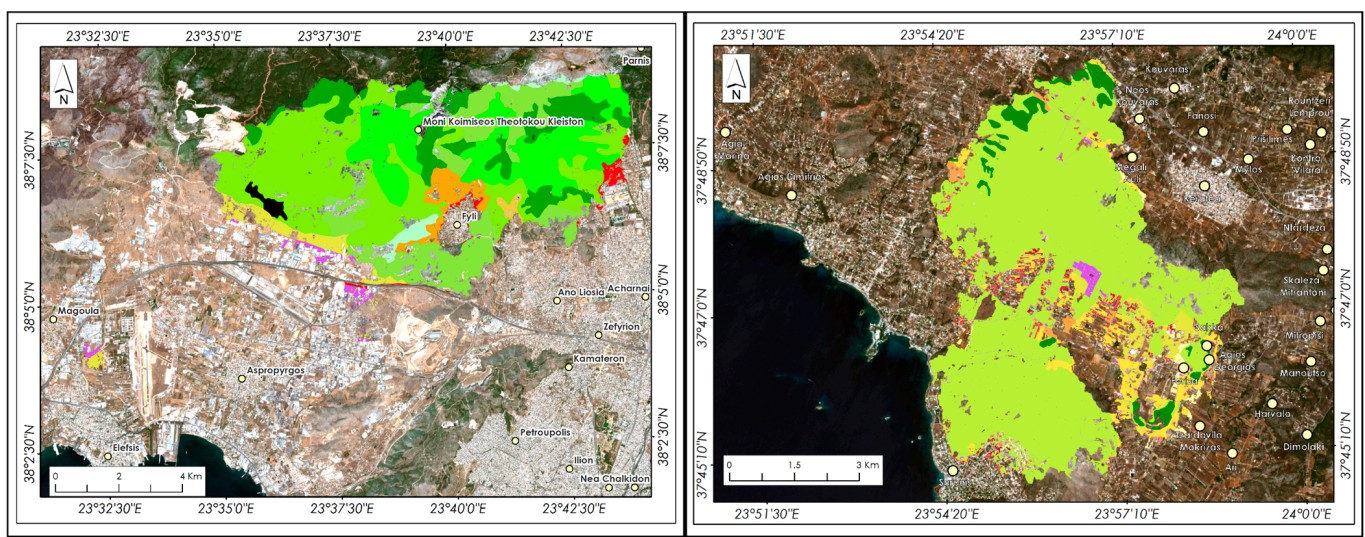

**Figure 17.** Parnitha–Lagonisi Land Cover of the Burned Area.

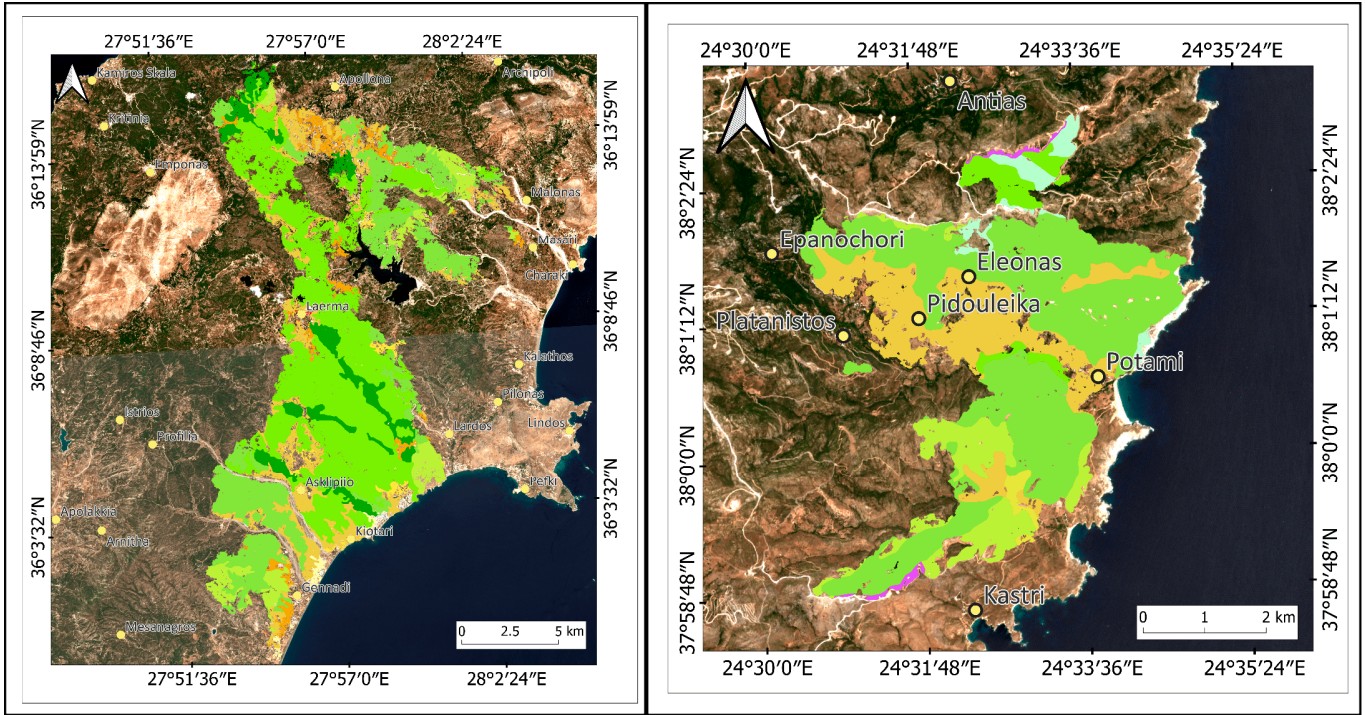

**Figure 18.** Rhodes–Evia CORINE Land Cover of the Burned Area.

**Table 10.** Dadia–Evros Land Cover Statistics.

| | Evros | |
|---|---|---|
| **CLC Code** | **Area (km$^2$)** | **Percentage (%)** |
| 112 | 0.22 | 0.02 |
| 121 | 2.64 | 0.29 |
| 122 | 2.22 | 0.25 |
| 131 | 0.55 | 0.06 |
| 211 | 62.53 | 6.92 |
| 212 | 1.05 | 0.12 |
| 223 | 7.19 | 0.80 |
| 231 | 2.71 | 0.30 |

**Table 10.** *Cont.*

| | Evros | |
|---|---|---|
| **CLC Code** | **Area (km²)** | **Percentage (%)** |
| 242 | 8.91 | 0.99 |
| 243 | 68.79 | 7.62 |
| 311 | 168.49 | 18.66 |
| 312 | 14.92 | 1.65 |
| 313 | 244.96 | 27.12 |
| 321 | 32.89 | 0.68 |
| 322 | 6.14 | 16.61 |
| 323 | 150.01 | 16.61 |
| 324 | 125.37 | 13.88 |
| 331 | 0.18 | 0.02 |
| 421 | 0.62 | 0.07 |
| 511 | 0.46 | 0.05 |
| 512 | 0.26 | 0.03 |
| 522 | 2.00 | 0.22 |

**Table 11.** Loutraki–Dervenochoria Land Cover Statistics.

| | Loutraki | | | Dervenochoria | |
|---|---|---|---|---|---|
| **CLC Code** | **Area (km²)** | **Percentage (%)** | **CLC Code** | **Area (km²)** | **Percentage (%)** |
| 122 | 0.04 | 0.37 | 112 | 0.07 | 0.06 |
| 142 | 0.13 | 1.20 | 121 | 0.01 | 0.01 |
| 223 | 0.03 | 0.27 | 122 | 0.04 | 0.37 |
| 242 | 0.58 | 4.57 | 131 | 0.72 | 0.65 |
| 243 | 1.90 | 17.41 | 211 | 0.06 | 0.05 |
| 312 | 1.27 | 11.59 | 223 | 0.82 | 0.74 |
| 323 | 1.36 | 12.46 | 231 | 1.25 | 1.12 |
| 324 | 5.71 | 52.16 | 242 | 2.20 | 1.98 |
| | | | 243 | 2.27 | 2.04 |
| | | | 312 | 24.59 | 22.06 |
| | | | 313 | 4.96 | 4.45 |
| | | | 321 | 4.28 | 3.84 |
| | | | 323 | 36.80 | 33.01 |
| | | | 324 | 31.84 | 28.56 |
| | | | 333 | 1.61 | 1.45 |
| | | | 512 | 0.00 | 0.00 |

**Table 12.** Parnitha–Lagonisi Land Cover Statistics.

| | Parnitha | | | Lagonisi | |
|---|---|---|---|---|---|
| **CLC Code** | **Area (km²)** | **Percentage (%)** | **UA Code** | **Area (km²)** | **Percentage (%)** |
| 112 | 0.58 | 1.01 | 11100 | 0.01 | 0.02 |
| 121 | 0.57 | 1.01 | 11210 | 0.16 | 0.44 |
| 122 | 0.07 | 0.13 | 11220 | 0.37 | 1.00 |
| 124 | 0.00 | 0.01 | 11230 | 0.16 | 0.44 |
| 131 | 0.01 | 0.01 | 11240 | 0.04 | 0.11 |
| 223 | 1.85 | 3.24 | 11300 | 0.27 | 0.74 |
| 231 | 1.92 | 3.37 | 12100 | 0.30 | 0.83 |
| 242 | 0.01 | 0.01 | 12220 | 0.56 | 1.54 |
| 243 | 0.31 | 0.54 | 13100 | 0.01 | 0.04 |
| 312 | 7.49 | 13.14 | 13400 | 0.03 | 0.07 |
| 313 | 11.88 | 20.85 | 14100 | 0.00 | 0.00 |
| 323 | 11.27 | 19.78 | 14200 | 0.01 | 0.02 |
| 324 | 19.55 | 34.30 | 21000 | 0.38 | 1.03 |

**Table 12.** *Cont.*

| | Parnitha | | | Lagonisi | |
|---|---|---|---|---|---|
| CLC Code | Area (km$^2$) | Percentage (%) | UA Code | Area (km$^2$) | Percentage (%) |
| 333 | 0.99 | 1.73 | 22000 | 0.32 | 0.88 |
| 334 | 0.49 | 0.86 | 23000 | 1.46 | 4.01 |
| | | | 24000 | 1.87 | 5.13 |
| | | | 31000 | 1.46 | 4.00 |
| | | | 32000 | 29.07 | 79.70 |

**Table 13.** Rhodes–Evia CORINE Land Cover Statistics.

| | Rhodes | | | Evia | |
|---|---|---|---|---|---|
| CLC Code | Area (km$^2$) | Percentage (%) | CLC Code | Area (km$^2$) | Percentage (%) |
| 112 | 0.06 | 0.03 | 121 | 0.19 | 0.96 |
| 142 | 0.26 | 0.15 | 231 | 0.23 | 1.16 |
| 211 | 1.29 | 0.75 | 243 | 4.46 | 22.54 |
| 222 | 0.01 | 0.01 | 321 | 2.25 | 11.37 |
| 223 | 6.77 | 3.92 | 323 | 10.95 | 55.33 |
| 242 | 2.10 | 1.22 | 324 | 0.99 | 5.00 |
| 243 | 20.63 | 11.94 | 333 | 0.72 | 3.64 |
| 312 | 14.04 | 8.13 | | | |
| 321 | 12.33 | 7.14 | | | |
| 323 | 38.50 | 22.29 | | | |
| 324 | 76.52 | 44.29 | | | |
| 331 | 0.24 | 0.14 | | | |

**Table 14.** Corfu–Magnesia CORINE Land Cover Statistics.

| | Corfu | | | Magnesia | |
|---|---|---|---|---|---|
| CLC Code | Area (km$^2$) | Percentage (%) | CLC Code | Area (km$^2$) | Percentage (%) |
| 142 | 0.00 | 0.00 | 112 | 0.05 | 0.07 |
| 223 | 0.02 | 0.08 | 121 | 0.39 | 0.48 |
| 242 | 1.08 | 4.95 | 122 | 0.41 | 0.50 |
| 243 | 0.39 | 1.78 | 131 | 0.26 | 0.31 |
| 313 | 0.00 | 0.00 | 132 | 0.05 | 0.06 |
| 321 | 2.51 | 11.55 | 133 | 0.26 | 0.31 |
| 323 | 4.60 | 21.20 | 211 | 20.00 | 24.27 |
| 333 | 13.11 | 60.39 | 212 | 4.94 | 6.00 |
| 523 | 0.01 | 0.04 | 222 | 3.00 | 3.64 |
| | | | 223 | 0.89 | 1.08 |
| | | | 231 | 3.16 | 3.84 |
| | | | 242 | 5.95 | 7.22 |
| | | | 243 | 4.26 | 5.17 |
| | | | 312 | 0.07 | 0.09 |
| | | | 321 | 9.02 | 10.94 |
| | | | 323 | 29.15 | 35.37 |
| | | | 411 | 0.01 | 0.01 |
| | | | 523 | 0.00 | 0.00 |

*3.3. Tree Cover Density of the Burned Area*

According to the tables for Tree Cover Density 2018 for the burned areas, the category with the highest percentage of covered areas is the first one—all non-tree-covered areas. These areas are regions with zero percentage of tree coverage. The non-tree category covered the 613.69 km$^2$ or 43.35% of the total burned area. For all the regions, the most affected category was the first one with no tree cover density. However, in most regions, there are categories of tree density that were destroyed by wildfires. More specifically, for

Loutraki (Figure 19) the most affected was the first category, with 6.35 km$^2$ or 58.05% and category 1–10%, with 1.32 km$^2$ or 12.09% (Table 15), while for Dervenochoria (Figure 19), the most affected was the first with 70.07 km$^2$ or 62.82% and the 40–50% category with 15.44 km$^2$ and 13.84% (Table 15). In Parnitha (Figure 20), the most affected was the first category, with 38.67 km$^2$ or 67.82% and the 50–60% category, with 6.21 km$^2$ or 10.89%. In Lagonisi (Figure 20) the most affected was the first category with 36.05 km$^2$ or 98.84% (Table 16); in Rhodes (Figure 21), the most affected was the first category with 131.33 km$^2$ or 76%, and after the 50–60% with 14.9 km$^2$ or 8.62%, in Platanistos (Evia) (Figure 21) was the first category with 19.636 km$^2$ or 99.2% (Table 17). In Corfu (Figure 22), the most affected was the first category with 16.75 km$^2$ or 77.03% and then the 50–60% category with 1.07 km$^2$ or 4.92%, in Magnesia (Figure 22), the most affected was the first category with 78.34 km$^2$ or 94.95% (Table 18) and also in Evros, it was the first category with 216.51 km$^2$ or 23.97% and then the 60–70% category with 152.27 km$^2$ or 16.86% (Figure 23, Table 19).

**Table 15.** Loutraki–Dervenochoria Tree Cover Density Statistics.

| Loutraki | | | Dervenochoria | | |
|---|---|---|---|---|---|
| **Tree Cover Density %** | **Area (km$^2$)** | **Area Percentage (%)** | **Tree Cover Density %** | **Area (km$^2$)** | **Area Percentage (%)** |
| All non-tree-covered areas | 6.35 | 58.05 | All non-tree-covered areas | 70.07 | 58.05 |
| 1–10% | 0.13 | 1.20 | 1–10% | 0.89 | 1.20 |
| 10–20% | 0.03 | 0.27 | 10–20% | 1.65 | 0.27 |
| 20–30% | 0.58 | 4.57 | 20–30% | 3.87 | 4.57 |
| 30–40% | 1.90 | 17.41 | 30–40% | 10.14 | 17.41 |
| 40–50% | 1.27 | 11.59 | 40–50% | 15.44 | 11.59 |
| 50–60% | 1.36 | 12.46 | 50–60% | 8.79 | 12.46 |
| 60–70% | 5.71 | 52.16 | 60–70% | 0.69 | 52.16 |
| 70–80% | 0.00 | 0.02 | 70–80% | 0.01 | 0.02 |
| 80–90% | 0.00 | 0.01 | 80–90% | 0.00 | 0.00 |
| 90–100% | 0.00 | 0.00 | 90–100% | 0.00 | 0.00 |

**Table 16.** Parnitha–Lagonisi Tree Cover Density Statistics.

| Parnitha | | | Lagonisi | | |
|---|---|---|---|---|---|
| **Tree Cover Density %** | **Area (km$^2$)** | **Area Percentage (%)** | **Tree Cover Density %** | **Area (km$^2$)** | **Area Percentage (%)** |
| All non-tree-covered areas | 38.67 | 67.82 | All non-tree-covered areas | 36.05 | 98.84 |
| 1–10% | 0.02 | 0.04 | 1–10% | 0.01 | 0.03 |
| 10–20% | 0.08 | 0.15 | 10–20% | 0.01 | 0.03 |
| 20–30% | 0.54 | 0.94 | 20–30% | 0.03 | 0.07 |
| 30–40% | 2.40 | 4.21 | 30–40% | 0.12 | 0.34 |
| 40–50% | 5.59 | 9.81 | 40–50% | 0.22 | 0.61 |
| 50–60% | 6.21 | 10.90 | 50–60% | 0.03 | 0.07 |
| 60–70% | 2.90 | 5.08 | 60–70% | 0.00 | 0.01 |
| 70–80% | 0.46 | 0.81 | 70–80% | 0.00 | 0.00 |
| 80–90% | 0.11 | 0.19 | 80–90% | 0.00 | 0.00 |
| 90–100% | 0.03 | 0.05 | 90–100% | 0.00 | 0.00 |

**Table 17.** Rhodes–Evia Tree Cover Density Statistics.

| | Rhodes | | | Evia | |
|---|---|---|---|---|---|
| **Tree Cover Density %** | **Area (km²)** | **Area Percentage (%)** | **Tree Cover Density %** | **Area (km²)** | **Area Percentage (%)** |
| All non-tree-covered areas | 131.33 | 76.00 | All non-tree-covered areas | 19.64 | 99.20 |
| 1–10% | 0.15 | 0.09 | 1–10% | 0.00 | 0.00 |
| 10–20% | 0.42 | 0.24 | 10–20% | 0.00 | 0.00 |
| 20–30% | 1.40 | 0.81 | 20–30% | 0.00 | 0.00 |
| 30–40% | 5.06 | 2.93 | 30–40% | 0.01 | 0.06 |
| 40–50% | 10.51 | 6.09 | 40–50% | 0.03 | 0.15 |
| 50–60% | 14.90 | 8.62 | 50–60% | 0.04 | 0.18 |
| 60–70% | 7.67 | 4.44 | 60–70% | 0.03 | 0.17 |
| 70–80% | 1.29 | 0.75 | 70–80% | 0.03 | 0.15 |
| 80–90% | 0.06 | 0.04 | 80–90% | 0.01 | 0.07 |
| 90–100% | 0.01 | 0.00 | 90–100% | 0.01 | 0.03 |

**Table 18.** Corfu–Magnesia Tree Cover Density Statistics.

| | Corfu | | | Magnesia | |
|---|---|---|---|---|---|
| **Tree Cover Density %** | **Area (km²)** | **Area Percentage (%)** | **Tree Cover Density %** | **Area (km²)** | **Area Percentage (%)** |
| All non-tree-covered areas | 16.75 | 77.03 | All non-tree-covered areas | 78.35 | 94.95 |
| 1–10% | 0.00 | 0.00 | 1–10% | 0.01 | 0.01 |
| 10–20% | 0.03 | 0.12 | 10–20% | 0.03 | 0.04 |
| 20–30% | 0.17 | 0.77 | 20–30% | 0.09 | 0.11 |
| 30–40% | 0.51 | 2.35 | 30–40% | 0.51 | 0.61 |
| 40–50% | 0.91 | 4.20 | 40–50% | 1.47 | 1.79 |
| 50–60% | 1.07 | 4.92 | 50–60% | 1.50 | 1.82 |
| 60–70% | 0.96 | 4.43 | 60–70% | 0.51 | 0.62 |
| 70–80% | 0.70 | 3.23 | 70–80% | 0.04 | 0.05 |
| 80–90% | 0.37 | 1.71 | 80–90% | 0.00 | 0.00 |
| 90–100% | 0.27 | 1.24 | 90–100% | 0.00 | 0.00 |

**Table 19.** Dadia–Evros Tree Cover Density Statistics.

| | Evros | |
|---|---|---|
| **Tree Cover Density %** | **Area (km²)** | **Area Percentage (%)** |
| All non-tree-covered areas | 216.51 | 23.97 |
| 1–10% | 0.26 | 0.03 |
| 10–20% | 2.78 | 0.31 |
| 20–30% | 17.67 | 1.96 |
| 30–40% | 54.89 | 6.08 |
| 40–50% | 88.50 | 9.80 |
| 50–60% | 116.18 | 12.87 |
| 60–70% | 152.27 | 16.86 |
| 70–80% | 147.35 | 16.32 |
| 80–90% | 81.32 | 9.01 |
| 90–100% | 25.32 | 2.80 |

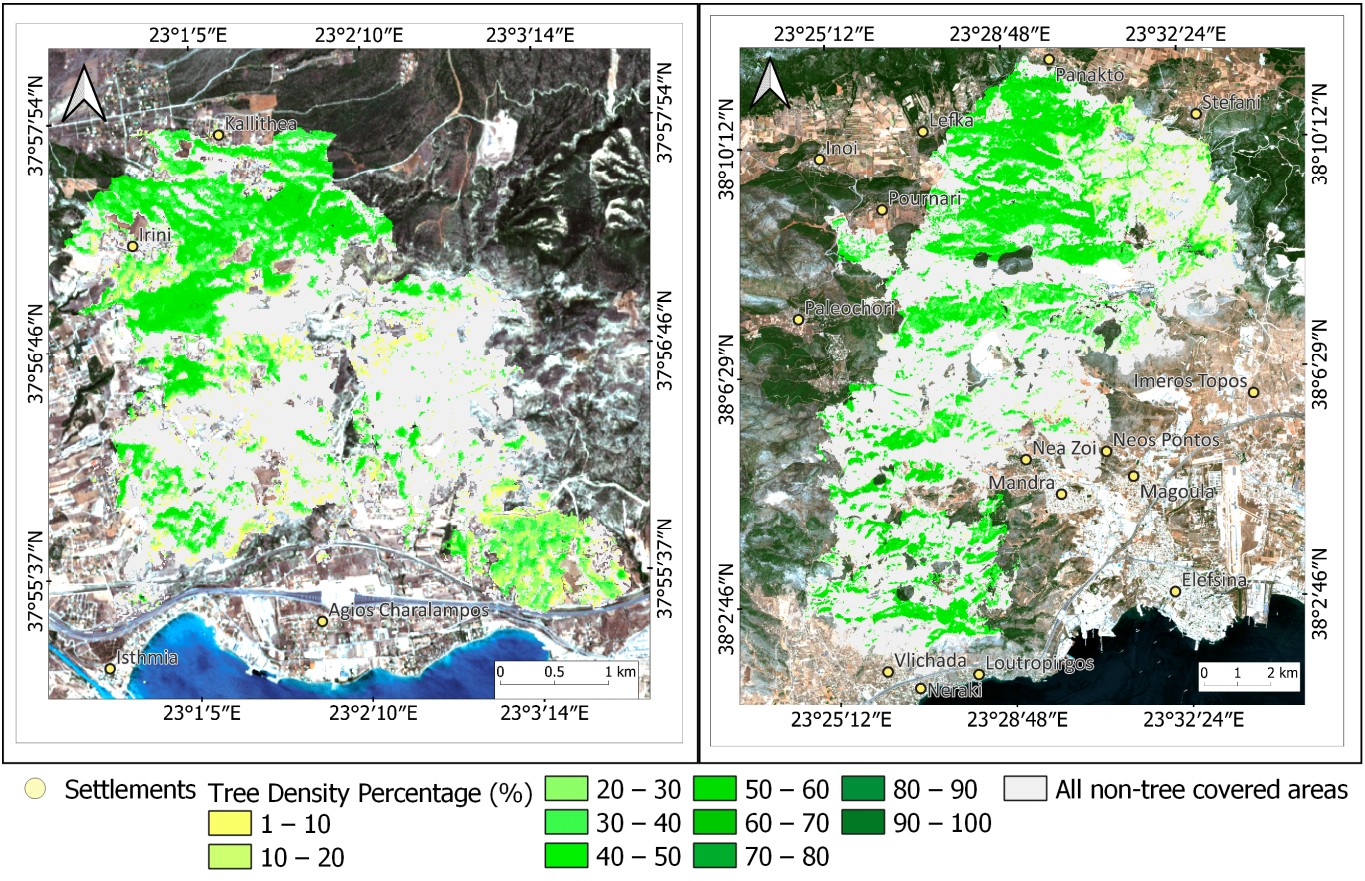

**Figure 19.** Loutraki–Dervenochoria Tree Cover Density of the Burned Area.

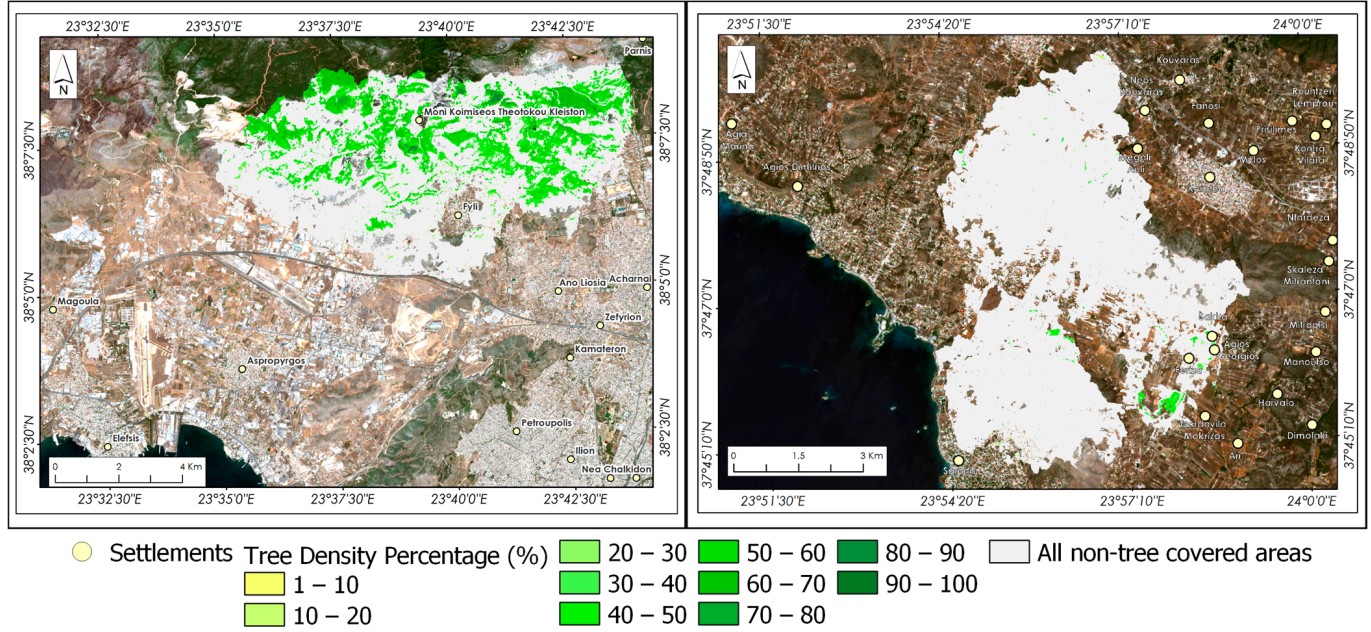

**Figure 20.** Parnitha–Lagonisi Tree Cover Density of the Burned Area.

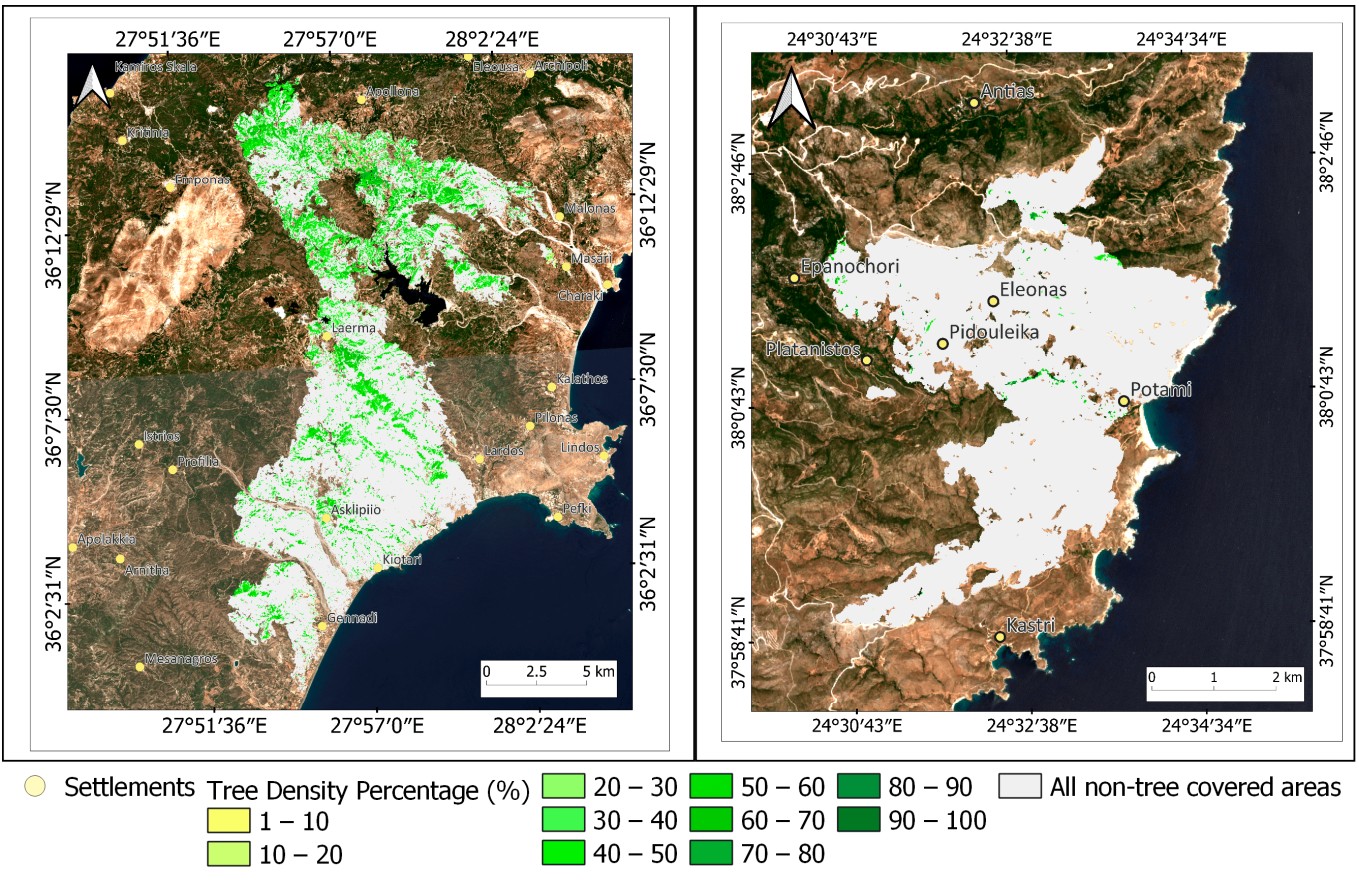

**Figure 21.** Rhodes–Evia Tree Cover Density of the Burned Area.

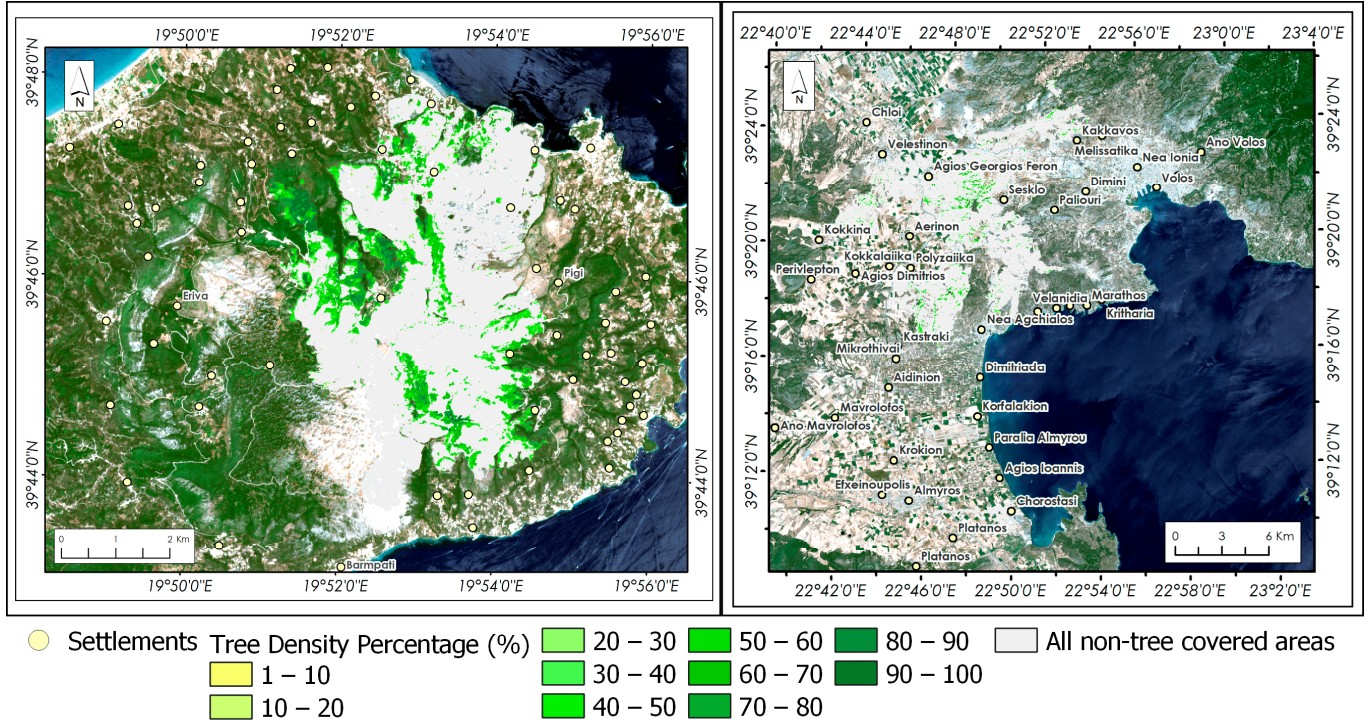

**Figure 22.** Corfu–Magnesia Tree Cover Density of the Burned Area.

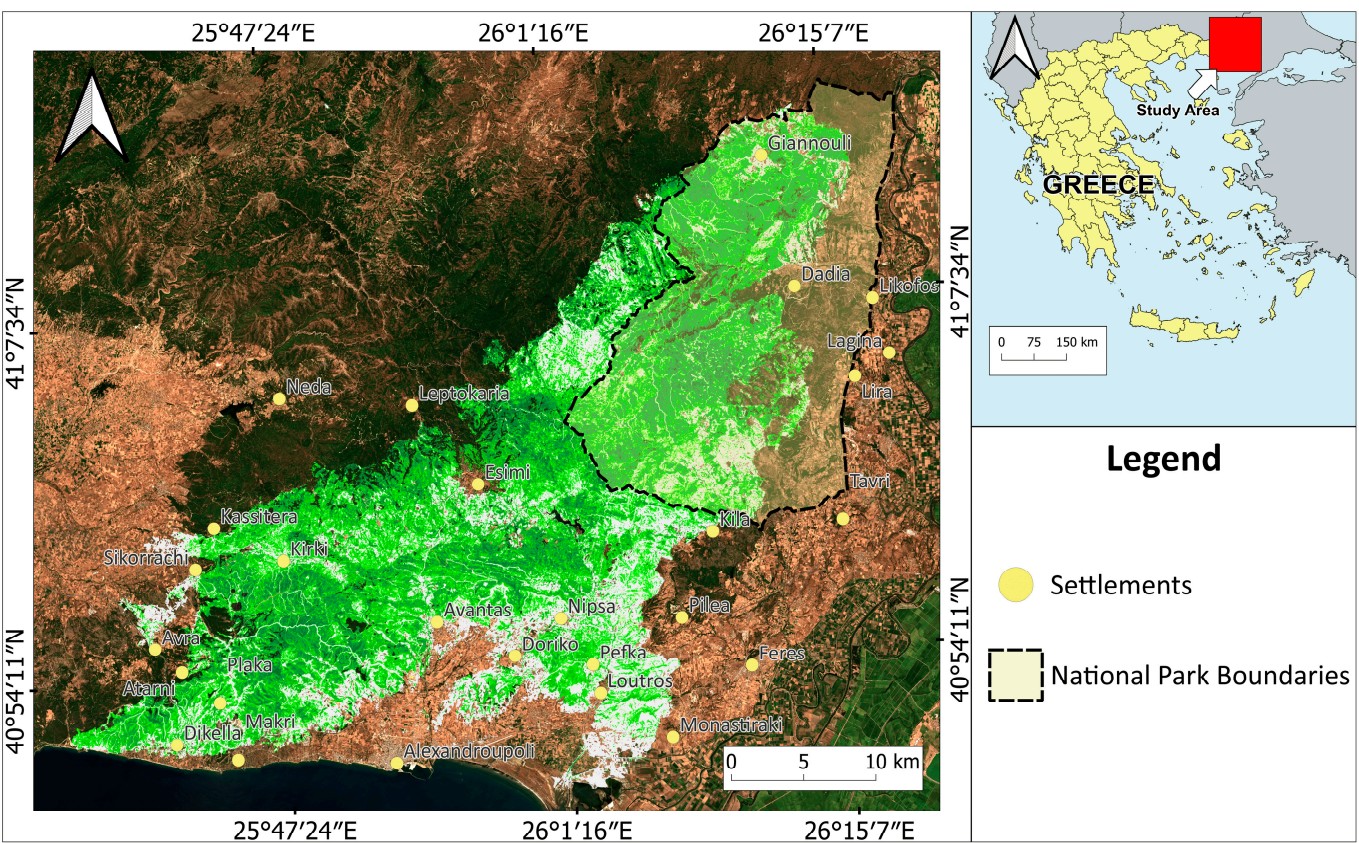

**Figure 23.** Dadia–Evros Tree Cover Density of the Burned Area.

*3.4. Validation of the Burned Area*

The validation of the burned areas is based on Copernicus Emergency Management Service–Mapping (CEMS–Mapping), which is a service of the EU's Copernicus program. CEMS–Mapping creates maps of natural disasters and emergencies and makes them available free of charge online. Some of these are fires, floods, humanitarian crises, landslides, and earthquakes. Two different types of maps are produced. Those that support the county's emergency management activities are produced within hours and days and combine four products. These products are a pre-event situation report, an initial damage assessment, a damage extent delineation, and a rating product that assesses the severity of the event. For this reason, these products are used to validate burnt areas in Greece. Secondly, the service also provides risk and recovery mapping for the prevention, preparation, and reduction of future events in damaged areas, as well as the assessment of the recovery phases [72].

According to Table 19, the Dervenochoria was calculated to have burned 111.53 km$^2$ as opposed to the 117.09 km$^2$ suggested by the Copernicus Land Service data. This provides a 95.25% accuracy rate for the burnt area assessed in the methodology. A similar accuracy rate of 94.26% was assessed in the Lagonisi–Kalivia fire, with an assessed burned area of 36.47 km$^2$ compared to the 38.69 km$^2$ of Copernicus' CLC 2018. The fire in Loutraki had a calculated burned area of 10.94 km$^2$, granting the lowest yet significant accuracy rate of 91.47% to Copernicus' 11.96 km$^2$. Additionally, the Rhodes forest fire's burned area was assessed with a very high accuracy ratio of 97.20% (172.77 km$^2$) compared to the burned area presented by the official Copernicus data (177.74 km$^2$). Northeastern Corfu had an affected area of 21.71 km$^2$ as opposed to the Copernicus data, which provided a burned area of 21.77 km$^2$, thus presenting an astonishing accuracy of 99.72%. The burned area of Evia's Platanistos region was assessed to be slightly bigger than Copernicus' 19.68 km$^2$, at 19.79 km$^2$, yet the difference is almost abysmal at 0.56% (0.11 km$^2$), hence granting the

assessment a 99.47% accuracy rate. The wildfire in Magnesia, Thessaly, was calculated to have affected 82.41 km$^2$ of the region. As opposed to this, Copernicus data suggest an area of 82.64 km$^2$. Thus, the assessment was quite accurate, at a 99.72% rate. The Dadia National Park's forest fire in Evros was the largest and most extensive one of those studied in this paper, calculating an overall burned area of 903.04 km$^2$. Compared to the data provided by Copernicus (938.81 km$^2$), the accuracy rate of the assessment was highly acceptable, at 96.19%, even though the difference in squared kilometers was quite high on a small scale. Last but not least, the region of Parnitha–Aspropyrgos in central Attica presented 56.98 km$^2$ of burned area in this study's results, at an accuracy rate of 91.99% against the burned area of 61.93 km$^2$ suggested in the validation data.

Finally, the total burned area of all the fires of summer 2023 in Greece presented in this study was assessed to be 1415.63 km$^2$. This correlates to 1470.31 km$^2$ in the burned area data from the Copernicus Land Service. As such, the overall accuracy rate of the methodology applied in this study was 96.28%. This accuracy rate is extremely high; hence, the results of the applied methodology were deemed acceptable.

## 4. Discussion

In light of the insights gathered from the study, several concluding remarks can be drawn. The integration of high-resolution datasets from the Copernicus Land Service and Sentinel-2 has significantly enhanced the precision and detail of wildfire impact assessments. The analysis of the nine major wildfires in Greece during the 2023 fire season underscores the escalating challenge of forest fires in the Mediterranean region, exacerbated by climate change and human activities. The employment of all the indices mentioned in this study has provided a nuanced understanding of burn severity, and they have proven to be reliable indicators of vegetation loss and landscape alteration. The study's reliance on these indices aligns with the work of previous researchers who have validated their effectiveness in assessing fire-affected areas.

The Dadia–Lefkimi–Soufli Forest National Park holds the distinction of being the first protected forested area in Greece. Its significance extends beyond conservation, as the park has been recognized as a key hub for ecotourism development since the late 1980s. Renowned for hosting remarkable European species, including endangered ones, the park has evolved into a crucial location in Greece. The imperative of safeguarding this ecosystem has been highlighted for decades.

The study's findings reveal a varied burn severity across the affected regions, with the Evros wildfire exhibiting the most extensive damage. More particularly, Evros presented a burn area of 903.04 km$^2$, making its extent 63.79% of the total burned area of summer 2023. Evros was followed by Rhodes and Dervenochoria with an area of 172.77 km$^2$ (12.20%) and 111.53 km$^2$ (7.88%) respectively. Magnesia was the fourth largest fire, 82.41 km$^2$, while the rest were below 30 km$^2$.

The spatial distribution of burn severity levels, predominantly moderate–high, indicates the necessity for targeted management and recovery efforts. Specifically, most of the total burned area presented moderate–high (33.87%) or moderate–low (29.39%) burn severity. High severity presented the lowest percentage of the total study area but was far from unimportant. Evros, being the largest fire of summer 2023, resulted in 337.40 km$^2$ being moderately to highly affected by the fire, making up 37.36% of Evros burned area and 23.83% of the total burned area. An area of 124.71 km$^2$ in Evros was of high severity, representing 13.81% of Evros' burned area and 8.81% of the total area.

The land cover analysis further highlights the significant effects on specific vegetation types, such as sclerophyllous vegetation and transitional woodland–shrub, which are more susceptible to high-severity burns. More analytically, in many regions, Transitional woodland–shrubs were the mainly affected vegetation type, such as in Loutraki (52.16%), in Parnitha (34.30%) and Rhodes (44.29%). Additionally, Sclerophyllous vegetation was the main burned land cover type in Dervenochoria (33.01%), Magnesia (35.37%), and Platanistos (55.33%). Lastly, Evros presented a massive burned area consisting mostly of

Mixed forests in a 244.96 km$^2$ extent. The remaining regions' main burned vegetation types include herbaceous vegetation or sparsely vegetated areas.

The comparisons of the general burned area of each wildfire suggest that the vast majority of the affected areas in all regions were not covered by trees, which is in line with the CLC data, which mostly represented shrubs and sclerophyllous plants. An important exception is the Evros fire, which affected about 16.86% of the area, presenting 60–70% tree coverage. Despite this, the majority of Evros burned area was devoid of trees, suggesting that the Mixed forests had partial tree coverage. Another impressive observation concerning the tree cover density in comparison to the burn severity of each region is that, while most of the total burned area lacks any tree coverage, in most areas, higher tree density seems to correlate directly with high burn severity. Despite this trend, Lagonisi, Platanistos, and Magnesia were clear exceptions. This was hardly surprising as these areas were almost entirely deprived of any trees whatsoever, making up 98.84%, 99.20% and 94.95% of their respective extent. All the results are closely tied to weather conditions during each fire incident compared to the average climate data for each affected area. Across all cases, the actual maximum daily temperature and wind speeds exceeded the averages. Humidity on fire days was higher than average in Corfu, Platanistos, Magnesia, and Evros, and lower in other areas.

The extensive environmental impacts of the Greek wildfires are profound and distressing. A significant part of the biodiversity in the affected areas has disappeared, including approximately 130,000 and 50,000 olive trees in Evros and Rhodes, respectively, and more than 1000 species of plants and animals in Parnitha. The fauna has also been severely affected, with thousands of beehives, mainly in Rhodes, and productive animals, especially in Magnesia, facing tragic losses. The rare black vulture of Evros, as well as the red deer and wolves of Parnitha, have been impacted. Additionally, numerous cats and dogs in private shelters in Attica perished in the flames. The wildfires in Magnesia destroyed 80% of the livestock, resulting in the tragic death of over 3000 animals. The impacts extend beyond animal casualties, affecting human life by disrupting vital ecological processes, highlighting the urgent need for effective forest and vegetation management to mitigate future risks.

The significance of open-source data in forest fire management and response is highlighted by this study, particularly through the enhanced capabilities provided by the Copernicus Land Monitoring Service. Providing free data encourages more people and organizations to participate in monitoring the environment and developing plans to reduce the damage caused by disasters. The validation process of burned areas, facilitated by the CEMS–Mapping, establishes a solid foundation for verifying the precision of the study's outcomes. The expedited delivery of maps and assessments has been proven crucial in confirming the severity of the wildfires. Nevertheless, it is important to acknowledge the inherent limitations of satellite data, such as the potential for cloud cover to conceal affected zones and the possibility that the resolution may not fully represent the extent of damage, especially in diverse landscapes. Furthermore, the study's findings must be extended to the broader socio-economic repercussions of the fires, encompassing the displacement of communities, the destruction of property, and enduring ecological disturbances.

As far as future research is concerned, it should focus on improving the methodologies used in this study, including the potential incorporation of a broader array of satellite datasets and more nuanced land cover classifications for even greater accuracy. Comparisons with other methodologies could also contribute to the validation process. There is also a critical need to advance the accuracy of burn severity indices by integrating satellite data with on-ground observations, creating a more holistic evaluation framework. Moreover, examining weather conditions in conjunction with climate data and defining biomass types will provide valuable indicators for pinpointing areas susceptible to fires. So, expanding the study to include more parameters could provide a more comprehensive understanding of wildfire impacts, which would be invaluable for future wildfire events.

The 2023 wildfires in Greece serve as a stark reminder of the increasing threat posed by wildfires across the Mediterranean region. These events highlight the urgent need to enhance monitoring capabilities and implement more effective management strategies. Specifically, the findings from this study contribute valuable insights that can assist not only Greek authorities but also the broader international community as they grapple with the global challenge of effectively managing wildfires in the context of climate change.

## 5. Conclusions

In general, the study on the 2023 wildfires in Greece emphasizes the growing importance of utilizing advanced satellite data and open-source information to better understand and manage wildfire events. The integration of various indices and datasets has proven valuable in assessing burn severity and vegetation loss while also highlighting the need for continuous improvement in methodologies and data sources. The study's findings contribute to the broader understanding of wildfire impacts in the Mediterranean region and beyond, emphasizing the urgency of addressing climate change and human activities that exacerbate these events. By refining monitoring and management strategies, the international community can work together to mitigate the effects of wildfires and protect vulnerable ecosystems and communities. Collaboration across scientific fields through an interdisciplinary approach can allow for cross-referencing findings from different fields and enable the development of integrated strategies to predict, prevent, and respond to wildfires more effectively.

**Author Contributions:** Conceptualization, I.P. and P.K.; methodology, A.D., E.K. and M.V.; software, A.D., E.K. and M.V.; validation, A.D., E.K. and M.V.; formal analysis, A.D., E.K. and M.V.; investigation, A.D., E.K. and M.V.; resources, A.D., E.K. and M.V.; data curation, A.D., E.K. and M.V.; writing—original draft preparation, A.D., I.A., E.K., M.V. and A.K.; writing—review and editing, A.D., I.A., E.K., M.V., A.K., P.K. and I.P.; visualization, A.D., E.K. and M.V.; supervision, I.P. and P.K.; project administration, A.D., P.K. and I.P. All authors have read and agreed to the published version of the manuscript.

**Funding:** This research received no external funding.

**Institutional Review Board Statement:** Not applicable.

**Informed Consent Statement:** Not applicable.

**Data Availability Statement:** Data are available upon request.

**Acknowledgments:** The Department of Geography, Harokopio University of Athens, provided the required facilities for this study, which the authors gratefully acknowledge. The authors are grateful to the European Space Agency, who provided Sentinel-2 data. The authors would also like to thank the reviewers for providing useful suggestions that enhanced the manuscript's quality.

**Conflicts of Interest:** The authors declare no conflicts of interest.

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
