# Peer review of "Employing Copernicus Land Service and Sentinel-2 Satellite Mission Data to Assess the Spatial Dynamics and Distribution of the Extreme Forest Fires of 2023 in Greece"

_fire, doi:10.3390/fire7010020_

Round 1
Reviewer 1 Report
Comments and Suggestions for Authors
This study assess the spatial dynamics and distribution of Greece's 2023 main forest fires. Experimental results show the good performance of the proposed method. However, there are still some issues that need further improvement in the article.
1) The introduction section needs further improvement. The author needs to introduce more updated methods and analyze the problems with the current methods. In addition, some image processing should also be introduced, such as:
[1] Super-resolution mapping based on spatial-spectral correlation for spectral imagery [J]. IEEE Transactions on Geoscience and Remote Sensing, 2021, 59(3): 2256-2268.
[2] A survey on object detection in optical remote sensing images[J]. ISPRS Journal of Photogrammetry and Remote Sensing, 2016, 117: 11-28.
2) The interpretability of each module designed by the authors is not strong, and the authors need to explain the interpretability of the designed module through formulas or physical limitations.
3) At present, the comparison methods in the article are relatively outdated. The author needs to add methods updated in the past three years for comparison.
4) Do the currently proposed methods still need some shortcomings that can be further improved in the future?
Author Response
Comments and Suggestions for Authors
This study assess the spatial dynamics and distribution of Greece's 2023 main forest fires. Experimental results show the good performance of the proposed method. However, there are still some issues that need further improvement in the article.
1) The introduction section needs further improvement. The author needs to introduce more updated methods and analyze the problems with the current methods. In addition, some image processing should also be introduced, such as:
[1] Super-resolution mapping based on spatial-spectral correlation for spectral imagery [J]. IEEE Transactions on Geoscience and Remote Sensing, 2021, 59(3): 2256-2268.
[2] A survey on object detection in optical remote sensing images[J]. ISPRS Journal of Photogrammetry and Remote Sensing, 2016, 117: 11-28.
Author’s response: Dear Reviewer thank you very much for your input, we appreciate your suggestions. We improved the introduction section and added some extra references. We fully respect your perspective regarding the incorporation of more contemporary methods. The primary aim of our article is not to introduce a groundbreaking method but rather to illustrate how the utilization of an established approach, in conjunction with various Copernicus products, yields the presented results. The comparison between the results obtained through Copernicus Emergency Management Service – Mapping and our calculations was not only necessary but also instrumental in ensuring the accurate evaluation and correction of our findings. Moreover, it is highly important the examination of the pre-fire tree density in the burned areas. It is observed a correlation – wherever the tree density was high, the burn severity increased accordingly.
2) The interpretability of each module designed by the authors is not strong, and the authors need to explain the interpretability of the designed module through formulas or physical limitations.
Author’s response: Dear Reviewer, thank you for taking the time to suggest improvements on the methodological description of our research, your input is greatly appreciated. We would like to note that we have provided formulas backed up by the related bibliography, which were utilized in the calculation of the various indices presented in this research. More particularly we have presented the equations suggested for the calculation of indices such as NBR, dNBR and RBR while employing preprocessed, atmospherically corrected satellite images, such as the ones we withdrew from Sentinel-2. We consider it imperative for the reader to have the tools necessary in replicating our methodological approach with similar data, thus we have emphasized the necessary parameters for these equations to function. We would be happy to consider any further input or specifications you may wish to share in further improving our manuscript.
3) At present, the comparison methods in the article are relatively outdated. The author needs to add methods updated in the past three years for comparison.
Author’s response: Dear Reviewer, thank you for your invaluable suggestions aimed at enhancing the quality of our article. We highly appreciate them. While we acknowledge that the method employed in our study may be somewhat outdated, it remains widely acceptable and still in use. Our primary intention is to combine the outcomes of this method, about the burned area and burn severity, with those obtained through Copernicus Emergency Management Service – Mapping. Furthermore, our article features the significance of pre-fire tree density and land cover in the areas under consideration. Your suggestion about incorporating new and established methods is duly noted for future comparisons between the results of each methodology.
4) Do the currently proposed methods still need some shortcomings that can be further improved in the future?
Author’s response: Dear Reviewer, we sincerely appreciate your insights for future improvement. In the future it would be beneficial for the scientific community to analyze those fire incidents using other methods and compare their respective outcomes. This approach will emphasize on the accuracy, advantages and disadvantages of each method, for a more comprehensive understanding. It will encourage other researchers to choose the most suitable method based on specific circumstances. Furthermore, we are considering the categorization of vegetation into biomass types, in order to explore its connection with burn severity and tree density of the affected areas. The analysis of the weather conditions in comparison with climate data, along with the definition of biomass types, can serve as valuable indicators for identifying fire-vulnerable areas. This can enhance protection measures and facilitate informed decision-making.

Reviewer 2 Report
Comments and Suggestions for Authors
Please see the attachment.

Moderate editing of English is required.
Author Response
Comments and Suggestions for Authors
Please see the attachment.
Comments on the Quality of English Language
Moderate editing of English is required.
Author’s response: Thank you very much for your suggestions. We revised accordingly.

Round 2
Reviewer 1 Report
Comments and Suggestions for Authors
Thank you for the authors‘ reply. I don't have any other questions.
Reviewer 2 Report
Comments and Suggestions for Authors
Dear Authors,
thank you for your careful revisions.
I think your study brings an interesting case from a recent fire.
Kind Regards,
Reviewer